# JoIN: Joint GANs Inversion for Intrinsic Image Decomposition

**Viraj Shah\***                                                                                      *vjshah3@illinois.edu*
*UIUC*

**Svetlana Lazebnik**                                                                                  *slazebni@illinois.edu*
*UIUC*

**Julien Philip**                                                                                      *julienov.philip@gmail.com*
*Adobe Research*

**Reviewed on OpenReview:** *https://openreview.net/forum?id=JEHIVfjmOf*

## Abstract

Intrinsic Image Decomposition (IID) is a challenging inverse problem that seeks to decompose a natural image into its underlying intrinsic components such as albedo and shading. While recent image decomposition methods rely on learning-based priors on these components, they often suffer from component cross-contamination owing to joint training of priors; or from Sim-to-Real gap since the priors trained on synthetic data are kept frozen during the inference on real images. In this work, we propose to solve the intrinsic image decomposition problem using a bank of Generative Adversarial Networks (GANs) as priors where each GAN is independently trained only on a single intrinsic component, providing stronger and more disentangled priors. At the core of our approach is the idea that the latent space of a GAN is a well-suited optimization domain to solve inverse problems. Given an input image, we propose to jointly invert the latent codes of a set of GANs and combine their outputs to reproduce the input. Contrary to all existing GAN inversion methods that are limited to inverting only a single GAN, our proposed approach, JoIN, is able to jointly invert multiple GANs using only a single image as supervision while still maintaining distribution priors of each intrinsic component. We show that our approach is modular, allowing various forward imaging models, and that it can successfully decompose both synthetic and real images. Further, taking inspiration from existing GAN inversion approaches, we allow for careful fine-tuning of the generator priors during the inference on real images. This way, our method is able to achieve excellent generalization on real images even though it uses only synthetic data to train the GAN priors. We demonstrate the success of our approach through exhaustive qualitative and quantitative evaluations and ablation studies on various datasets.

## 1 Introduction

Any natural image can be seen as an intricate combination of geometry-dependent, material-dependent, and lighting-dependent components. Decomposing an image into such intrinsic components is highly useful, since it allows for varying each intrinsic component independently in order to achieve complex image editing such as image relighting, material swap, and object insertion (Bousseau et al., 2009).

In a commonly used image decomposition framework, an image is typically decomposed into a light independent component referred to as albedo, a light dependent component referred to as shading, and optionally

---

\*This work was done during an internship at Adobe Research.
 Project page: **https://virajshah.com/join/**

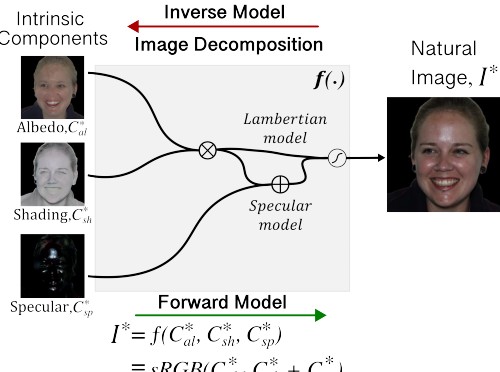

Figure 1: **Intrinsic Image Decomposition (IID) framework.** We consider a commonly used image decomposition framework that aims at decomposing the natural image into its light independent (albedo), light dependent (shading), and optionally residual (specular) components. The forward model $f(\cdot)$ from the intrinsic components to the natural image is simply given by multiplication of albedo and shading with addition of specular as a residual (here, $sRGB(\cdot)$ indicates the standard tone-mapping operation). However, the inverse mapping of natural image to its intrinsic components is highly ill-posed inverse problem that we aim to solve.

a residual component referred to as specular which captures the non-ideal behavior of natural surfaces. As depicted in Fig. 1, the forward mapping $f(\cdot)$ from the decomposed image components to the natural image in this case is relatively straightforward to obtain. However, the inverse mapping of the natural image to its intrinsic components is highly challenging ill-posed inverse problem owing to potentially infinite solution space. To obtain useful decomposition, it is required to constrain the solution space by employing priors on the image components.

Existing IID approaches leverage variety of hand-crafted or learned priors for recovery of different intrinsic components. However, priors from both of these categories often suffer from insufficient disentanglement and cross-contamination of components, *i.e.* features from shading may appear in albedo and vice versa. This effect is more pronounced in learning-based approaches that attempt to train a single model to predict multiple image components jointly (Das et al., 2022a; Yu & Smith, 2021). Further, in the case of learning-based priors, collecting the training data with groundtruth albedo and shading of natural images is extremely difficult, thus, they are typically trained using the simulated/synthetic data. Priors trained on simulated data are typically kept frozen during inference on real images, thus, they lead to poor generalization on real scenes which is called Sim-to-Real gap for IID tasks. In this work, we propose to solve both the challenges by using bank of Generative Adversarial Networks (GANs) as priors where each GAN is trained independently only on a single image component, thus alleviating the challenge of cross-contamination. Moreover, even though our GANs are trained on synthetic data, we allow for careful fine-tuning of the generator priors for inference on real scenes to bridge the Sim-to-Real gap.

The idea of using a GAN model as prior is explored widely for a variety of inverse imaging problems such as image inpainting (Yeh et al., 2016; Bora et al., 2017; Gu et al., 2019), HDR imaging (Niu et al., 2020), image super-resolution (Ledig et al., 2016; Bora et al., 2017; Gu et al., 2019), compressive sensing (Bora et al., 2017; Shah & Hegde, 2018a), and phase retrieval (Hand et al., 2018; Hyder et al., 2019), owing to their tremendous success in mapping a lower-dimensional latent distribution to the manifold of real images. Such approaches assume that the target image lies on the range of the GAN and attempt to obtain the corresponding latent code through optimization. However, unlike the imaging problems where the solution set consists of only a single natural image, in case of IID, the solution consists of several intrinsic components. For that reason, when used for IID task, such GAN prior is required to generate all the decomposition components from a single latent code. For example, in a previous work (Guo et al., 2020) all intrinsic components are generated simultaneously by a single GAN. Such a design is susceptible to the problems of poor disentanglement and cross-contamination as all components are generated by a single model and tied with a single latent code.

To this end, in this work we propose to use a bank of independently trained generators as a prior: we train three separate GANs – one each for albedo, shading, and specular. We model the natural image as a combination of the intrinsic components generated by these component-specific GANs, and find the latent codes for the albedo, shading, and specular GANs such that when input to the corresponding GAN model, they accurately generate their respective intrinsic component, and produce the input image when combined (Fig. 2). Since each GAN is trained independently on only a single image component, each generator has no capability of generating anything else other than the image component on which it is trained. Therefore, an independently trained GAN provides a much stronger constraint on the solution space as compared to the existing methods.

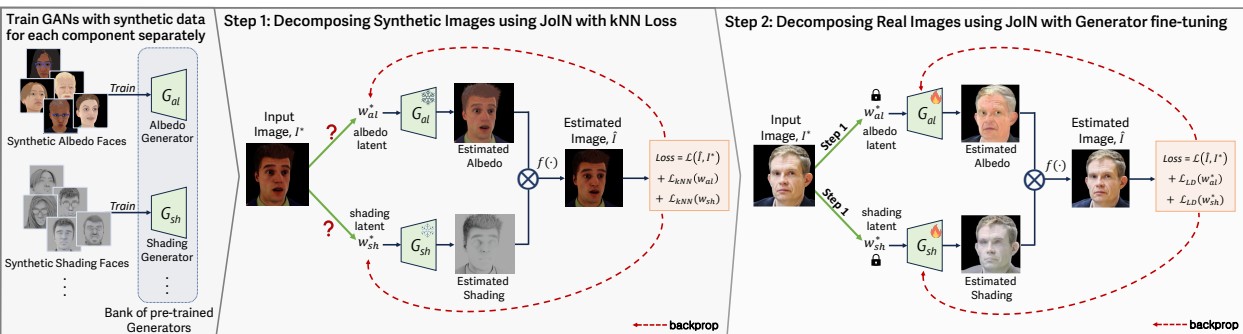

Figure 2: **Overview of our approach. Left:** We use a bank of pre-trained GANs as a prior where each GAN is trained only on a single image component using synthetic data. We design the problem of decomposition as a joint GAN inversion problem on multiple GANs using the input image $I^*$ as the only supervision. **Step 1:** We aim to optimize the latent codes of each GAN in a way that after passing the outputs of individual GANs through forward mapping $f(\cdot)$, the resulting image estimate $\hat{I}$ resembles the input image $I^*$. Apart from using reconstruction loss, we propose to use kNN loss on the latent codes to strongly enforce the priors learnt by each GAN. Such approach leads to successful decomposition on synthetic data. **Step 2:** For decomposition on real images, we bridge the Sim-to-Real gap by *carefully* fine-tuning the individual GANs in a way that they can represent the real image features while still maintaining strong component-wise priors.

Using more than one GAN as a prior comes with new set of challenges. Existing approaches restrict the solution space by inverting the mapping of a pre-trained GAN, *i.e.* finding the latent code corresponding to the target image. Inverting the mapping of a GAN is a highly challenging and well-studied problem known as GAN Inversion. In our case, we need to obtain the latent codes for all the GANs in our bank – meaning inverting multiple GANs simultaneously. Since all the existing GAN inversion methods are limited to inverting only a single GAN, it is not straightforward to use them to jointly invert all the GANs in our bank. To this end, we propose a new Joint GAN Inversion algorithm, JoIN, that can jointly invert multiple GANs without requiring any component-wise supervision (see Fig. 2). To the best of our knowledge, JoIN is the first method to demonstrate successful inversion of multiple GANs at once.

Key to successful joint inversion is our kNN-based loss regularization that ensures the latent code estimates for each intrinsic component remains within the distribution manifold of its associated GAN. Further, we use only the synthetic data to train our GANs, and show that our approach successfully generalizes to real images by adapting generator fine-tuning technique from the existing GAN inversion literature in our pipeline: unlike the existing learning-based approaches that keep the priors frozen during the inference, we allow for a careful fine-tuning of our generator priors to bridge the Sim-to-Real gap. We evaluate our approach on different datasets such as faces, materials, and indoor scenes, and provide extensive comparisons on both the synthetic and real images.

Since we train a separate GAN model for each image component, our approach remains modular with respect to the choice of forward model, *i.e.* if we choose to add a new component (*e.g.* specular) to our existing forward model, we can reuse the GANs trained for existing components (albedo and shading) in our bank, and need to train only one GAN from scratch. We demonstrate such capability along with other ablation studies in our experiments.

To summarize, our key contributions are following:

- We propose to use a bank of independently trained GANs as priors to tackle the IID task. Training each GAN separately for individual intrinsic components provides much stronger priors and mitigates the challenges of poor disentanglement and signal cross-contamination.

- We propose a new Joint GAN Inversion algorithm, JoIN, that can jointly invert multiple GANs without requiring any component-wise supervision (see Fig. 2(step 1)). Our method decomposes an image by successfully inverting it onto albedo GAN, shading GAN, and specular GAN simultaneously using only the natural image as a supervision. To the best of knowledge, ours is the first method to successfully invert multiple GANs at once.

- We introduce a novel kNN-based regularization term for joint GAN inversion allowing to better retain their distribution properties of each GAN as compared to existing regularizations.

- We show that even though the bank of GANs is trained entirely on synthetic data, our approach can achieve excellent generalization on the real images by leveraging existing GAN inversion techniques such as generator fine-tuning (see Fig. 2(step 2)). Our model also offers flexibility of decomposing into either two components (albedo, shading) or three components (albedo, shading, specular).

## 2 Related Work

### 2.1 Image Decomposition

The intrinsic decomposition of images is a long-standing problem that emerged with the retinex and lightness theory (Land & McCann, 1971; Horn, 1974). Using a purely diffuse model, early work computed shadow-free reflectance images of faces (Weiss, 2001) using time-lapse sequences. User-assisted methods were the first to successfully tackle the problem for single natural images (Tappen et al., 2003; Bousseau et al., 2009). With the rise of deep learning a large body of work emerged using neural networks to predict intrinsic images (Zhou et al., 2015; Nestmeyer & Gehler, 2017; Li & Snavely, 2018b;a; Yu & Smith, 2019; Li et al., 2020). While feed-forward methods such as (Das et al., 2022b;a; Yu & Smith, 2021) bring improvements in terms of model architecture and loss functions, they are prone to cross-contamination since they use a single model for predicting multiple components. Recently, CRefNet (Luo et al., 2023) proposed decoder-sharing transformer architecture for albedo prediction, but their model is limited to only a single component unlike our approach that can provide full decomposition of an image into 2 or 3 components. Similarly, S-Aware (Jin et al., 2023) too is limited to predicting only the albedo. Careaga & Aksoy (2023) predicts albedo and shading using two stage networks, but requires dense pseudo ground truth for generalization to real images. Liu et al. (2020) presented a method that shares some ideas with our pipeline: they build priors independently for shading and reflectance using image collections and adversarial auto-encoders. Contrary to our approach, all such feed-forward methods use a fixed forward model and would require full retraining if a component such as specular were to be added. One can refer to a recent review (Liu et al., 2024) for a summary of the research in intrinsic decomposition.

For photo collections, many works explored this problem using geometry priors (Laffont et al., 2012; 2013; Duchêne et al., 2015; Philip et al., 2019; 2021), and recently methods were introduced working on videos (Ye et al., 2014; Bonneel et al., 2014; Meka et al., 2016; 2021). One of the main applications of IID lies in image relighting for which decomposing an image in specific components such as albedo, shading, shadows or normals is often an intermediate step both for faces (Pandey et al., 2021; Hou et al., 2021; Yeh et al., 2022) and outdoor picture (Griffiths et al., 2022). A recent work (Zhang et al., 2024) trains an end-to-end image relighting model by encoding the source scene and target illumination into a latent space, and recovers the albedo maps of the scene from its latent representation. However, the quality of intrinsic components obtained by such methods is generally poor since their primary aim is relighting, while intrinsic decomposition is treated as byproduct. In the context of the more constrained problem of material acquisition, methods often aim at recovering richer information. They usually predict a full SVBRDF map composed of albedo but also normals, roughness, and specular maps (Aittala et al., 2015; Deschaintre et al., 2018; 2019). Some recent methods proposed to use adversarial networks as part of their SVBRDF estimation pipelines (Zhou & Kalantari, 2021; Zhou & Khademi Kalantari, 2022). Closer in spirit to our method MaterialGAN (Guo et al., 2020) proposes to use a single GAN trained on all the channels of SVBRDF maps and to invert a latent code to recover an SVBRDF. Contrary to our method it requires several input images and known lighting conditions for each input.

Leveraging different learning-based priors for different signal components has been explored in previous works such as DoubleDIP (Gandelsman et al., 2019) and its extensions (Ren et al., 2020; Zhou et al., 2023; Zhuang et al., 2022) that use one untrained deep image prior (DIP) for each signal component. Their approach relies on strong internal self-similarity property, i.e., the patches within each component have higher self-similarity than the patches in the combined image, while the patches across the two components are highly dissimilar. This strategy has proven effective in tasks like image dehazing, layer separation, and watermark removal,

where the components are often structurally dissimilar. However, the assumption of structural dissimilarity of the components may not hold for albedo and shading, since their features (shape, edges) and structure are highly similar to each other. Evidently, UIDNet (Zhang et al., 2022) demonstrated that applying DoubleDIP to IID results in severe cross-contamination and a loss of fine details. They introduce additional hand-crafted priors, and use encoder-decoder with skip connections to mitigate these issues. Using untrained DIP priors is computationally expensive with inference times of several minutes. In our case, instead of untrained DIPs, we employ multiple trained GANs that provide highly accurate priors, and by inverting them jointly, we effectively avoid the cross-contamination, preserve details, and achieve significantly faster inference times.

## 2.2 GAN Priors and GAN Inversion

Inverting a GAN to obtain the latent code corresponding to an image is a key step in solving inverse problems (Shah & Hegde, 2018b). Owing to its importance, many GAN inversion approaches have been recently proposed as reviewed in Xia et al. (2021). We can classify these methods in three broad classes: optimization-based, encoder-based, and hybrid.

**Optimization-based methods** such as Lipton & Tripathi (2017b); Creswell & Bharath (2019); Zhu et al. (2016); Lipton & Tripathi (2017a); Karras et al. (2020b); Abdal et al. (2019; 2020) regress a latent code using gradient descent by minimizing a loss between a target image and the generated image produced by the latent code. The key variations among these approaches are the latent space chosen for the optimization ($\mathcal{Z}, \mathcal{W}$, and $\mathcal{W}^+$ space); the loss function(s) they use to calculate the similarity between the target and the estimate; and additional criterion used for aiding optimization.(such as regularization, improved initialization, and early stopping). Although these methods can produce relatively precise outcomes, their iterative optimization scheme is slower than forward methods and can converge to local minima when badly initialized.

Contrary to optimization-based approaches, **encoder-based methods** propose to use an end-to-end framework based in which an encoder that takes the target image as input, predicts the latent code directly (Xia et al., 2021; Pidhorskyi et al., 2020; Richardson et al., 2021a; Alaluf et al., 2021; Chai et al., 2021a; Larsen et al., 2016; Zhu et al., 2016; Brock et al., 2016; Perarnau et al., 2016; Tov et al., 2021; Wei et al., 2021; Wang et al., 2021; Yao et al., 2022; Moon & Park, 2022; Mao et al., 2022). Encoder-based methods have faster inference times, but they suffer from lower result quality and poorer generalization. They can be used as initializer for optimization-based inversion. Methods that combine both optimization and encoder-based initialization are known as **hybrid approach** (Chai et al., 2021b; Bau et al., 2019c; Zhu et al., 2016; Alaluf et al., 2021; Bau et al., 2019b;a; Huh et al., 2020; Wei et al., 2021).

Recently, a fourth category of approaches has emerged that advocates fine-tuning the generator weights after optimizing the latent code (Roich et al., 2021; Feng et al., 2022). Such methods produce better reconstruction since the generator is allowed to fit the target image but fine-tuning the generator can lead to distortions in the generator manifold losing important properties. More recently, (Bhattad et al., 2023) has shown that StyleGAN has an internal representation of intrinsic images that can be accessed by finding an appropriate offset to the latent codes. However, this method is not applicable for real images, since the offsets obtained are not compatible with the inverted latent codes of the real images.

In recent works, a new category of generative models known as diffusion models (Dhariwal & Nichol, 2021) are becoming increasingly popular in many application domains owing to their superior performance in generating high-quality and diverse images. Several works (Feng et al., 2023; Li et al., 2023) leverage diffusion models as priors in solving inverse problems such as reconstruction and super-resolution. Similar in spirit to our method, the concurrent work Wang et al. (2024) has explored using independently trained diffusion models for blind image deblurring, where a blurry image is decomposed into a clean image, blur kernel, and a tilt map. Note that unlike the IID, here the target components are structurally *incoherent* and dissimilar, e.g. the blur kernel is typically much lower dimensional and exhibit distinctly different features than the natural image. Moreover, Diffusion models are computationally more expensive than GANs during both training and inference, and does not allow for smooth transitions in latent space (that can be leveraged for image relighting in our case). While we acknowledge leveraging multiple diffusion priors for the IID task is a significant direction for future research, especially to model larger and more diverse datasets, our

work demonstrates that GANs offer a powerful and efficient tool for IID, particularly when computational resources and inference speed are important considerations.

## 3 Our Approach

In this work, we propose to solve intrinsic image decomposition using an optimization formulation that leverages a bank of GANs as priors. We use the fact that each image component distribution can be closely approximated using a GAN model, and create a bank of GANs by training a separate GAN model for each image component that we desire to recover (See Fig. 2(left)). Since each GAN maps a lower dimensional latent space to an image component space, solving the inverse problem aims to recover the latent codes corresponding to each image component which can be formulated as the joint inversion of multiple GANs as shown in Fig. 2(step 1).

In this section, we first present our optimization framework and its constraints, then we introduce a novel loss that helps further regularize the optimization and maintain the GANs priors.

### 3.1 JoIN: Joint Inversion of Multiple GANs

Given an image $I^*$, and a forward differentiable function $f$ taking as input $n$ image components $C^*_{1\ldots n}$ we suppose that $I^*$ can be modeled as:

$$I^* = f\left(C^*_1, \ldots, C^*_n\right) \tag{1}$$

Where we assume $f$ is known. We aim at recovering $C^*_{1\ldots n}$ given only $I^*$ as input. Instead of directly optimizing $C_{1\ldots n}$ to recover $C^*_{1\ldots n}$ in the image domain we propose to represent them as the output of a GAN $G_i$ for a latent code $w_i$:

$$C_i = T_i^{-1}\left(G_i\left(w_i\right)\right) \tag{2}$$

Where $T_i^{-1}$ is an inverse color transform operator.

For $G_i$ we follow the architecture of StyleGAN2 (Karras et al., 2020a), while $w_i$ refers to an intermediate latent code in $\mathcal{W}$ space. Note that the StyleGAN-based architecture first maps a Gaussian distribution ($\mathcal{Z}$) to an intermediate latent space $\mathcal{W}$ using a non-linear mapping network, and $\mathcal{W}-$space is then mapped to distribution of real images. $\mathcal{W}-$space is known to provide higher degree of disentanglement and smoother control over various image features (Karras et al., 2019b), thus has emerged as more suitable choice for performing GAN inversion (Xia et al., 2021). We follow the same and perform the joint inversion in $\mathcal{W}-$space. Our estimate of $I$ is thus given by:

$$I(w_1, \ldots, w_n) = f\left(T_1^{-1}\left(G_1\left(w_1\right)\right), \ldots, T_n^{-1}\left(G_n\left(w_n\right)\right)\right) \tag{3}$$

Our goal is thus to find the $w_i, i \in 1\ldots n$ that minimize a reconstruction loss function $\mathcal{L}$:

$$\underset{w_i, i\in 1\ldots n}{\arg\min} \mathcal{L}\left(I^*, I(w_1, \ldots, w_n)\right). \tag{4}$$

Eq. 4 is analogous to the standard GAN inversion formulation, except for the fact that it aims to invert $n$ Generators simultaneously instead of one. The loss in Eq. 4 is minimized over all the latent codes $w_i$s simultaneously, and the final estimates $\hat{w}_i$s can be passed to Eq. 2 to obtain the image components $C^*_{1\ldots n}$. We use E-LPIPS loss (Kettunen et al., 2019) (robust version of LPIPS (Zhang et al., 2018) perceptual loss) as the loss function $\mathcal{L}$ unless otherwise specified.

### 3.2 Intrinsic Image Decomposition

In this paper, we focus on the problem of intrinsic image decomposition, covering several formulations (Garces et al., 2021). We thus adapt the general formulation presented in the previous section.

We first consider a simple forward model assuming Lambertian surfaces (Barrow & Tenenbaum, 1978). This seminal model uses two components – albedo and shading to represent the image:

$$I^* = sRGB(C_{albedo} \cdot C_{shading}). \tag{5}$$

Then we consider a non-Lambertian model with a separate specular component:

$$I^* = sRGB(C_{albedo} \cdot C_{shading} + C_{specular}). \tag{6}$$

This model merges the specular color and specular illumination of the dichromatic model Tominaga (1994) as a single component.

### 3.3 Optimization-based Joint Inversion

Given a forward model described in Eq. 5 or Eq. 6, we first train a GAN for each intrinsic component (albedo, shading and eventually specular images) independently using the StyleGAN2 architecture (Karras et al., 2019b; 2020b). As the GANs are trained independently the forward model can be chosen after the fact and components can be replaced or added.

Given the pre-trained GANs and an input $I^*$, we then optimize latent codes $w_i^*$ by minimizing the loss function in Eq. 4 over $w_i$s using gradient descent. Note that our optimization is only guided by the input image $I^*$ since the loss in Eq. 4 is calculated on the combination of all the estimated components.

### 3.4 kNN Loss

Allowing the latent codes $w_i$ to explore the latent space freely during optimization can lead to cross-contamination as they deviate from the manifold of possible $w$ within $\mathcal{W}$. In such case the generator for a component $C_i$ starts producing signals corresponding to another component $C_j$. We thus propose a new kNN-based regularization loss to constrain the latent codes to stay within domain.

To this end, we take inspiration from previous works (Zhu et al., 2020; Tov et al., 2021), which introduces an in-domain loss on the latent codes as follows:

$$\mathcal{L}_{in} = \|w_i - \bar{w}\|_2, \tag{7}$$

where $\bar{w}$ is the average $w$ obtained by drawing large number of random samples (typically $100k$) of $w-$codes from $\mathcal{W}$ space. This in-domain loss tries to keep the optimized latent close to the distribution mean in order to keep the estimate within the training domain. We argue that this formulation has two main drawbacks. First, using the distribution mean penalizes the solution space near the boundary of the distribution.

Second, it assumes that the distribution of $w$ is isotropic, which has no reason to hold given the use of a non-linear mapping network that maps the Gaussian distribution $\mathcal{Z}$ with the intermediate latent distribution $\mathcal{W}$ during training of StyleGAN model. We depict such limitation of in-domain loss in Fig. 3 where we first plot the distribution of $w-$codes by projecting $\mathcal{W}-$space in two dimensions using t-SNE (van der Maaten & Hinton, 2008), and validate the non-isometric behavior of $\mathcal{W}-$space. Since the distribution varies differently along different directions, one can see that simply using the

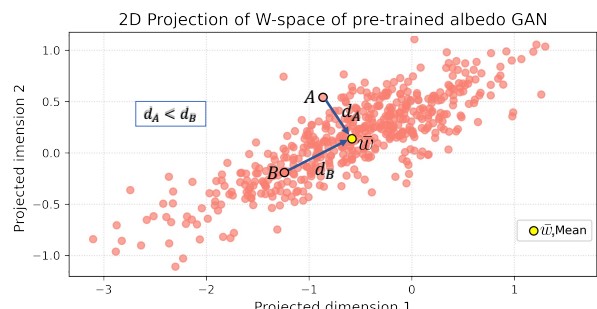

Figure 3: $\mathcal{W}-$**space of a pre-trained GAN is non-isometric.** The t-SNE 2D projection of $\mathcal{W}-$space of pre-trained albedo GAN confirms its non-isometric behavior as the distribution varies differently across the dimensions. Naively using the distance from the mean as a loss penalizes point $B$ more than point $A$, even though $B$ is well within the distribution unlike $A$. As a remedy, our proposed kNN loss uses the closeness of a point to its neighbors as a loss instead of relying on its distance from the distribution mean, promoting exploration of the entire space.

Figure 4: **Comparison: in-domain vs. kNN loss. Left:** In-domain loss attracts the estimate towards the center resulting in higher loss near the distribution boundary. **Right:** kNN loss ($k = 5$ NN) attracts the estimate towards the nearby latents, allowing to better capture the diversity of the distribution while still keeping the estimate within the distribution. Dots represent randomly sampled latent vectors (=100) and the colors map the loss value.

distance from the mean $\bar{w}$ as a loss (as done in case of in-domain loss) would penalize point $B$ more than point $A$ since $d_B > d_A$, even though point $B$ is well within the distribution while $A$ lies outside. Note that we use randomly sampled $100k$ $w-$codes to calculate the t-SNE projection and the distribution mean $\bar{w}$, while the plot shows randomly selected 1000 of them for legible visualization.

As a remedy to both the issues, we propose a novel k-Nearest Neighbors-based loss formulation that better allows the exploration of the full domain while still ensuring the estimate remains in-distribution. Instead of pushing the estimate $w_i$ towards the mean, we steer it closer to its neighboring $w$ codes to ensure it remains within the distribution. In non-isometric distributions like the one in Fig. 3, proximity to other samples is a more reliable indicator of a point belonging to the distribution than its distance from the mean. This approach allows the estimate to move freely within the distribution, rather than being constrained to the mean, and avoids unfairly penalizing estimates near the boundaries. We first sample a large number $(100,000)$ of $z$ codes from a truncated Gaussian distribution, and pass them through the mapping network of StyleGAN2 to obtain a set $\mathcal{S}_{\mathcal{W}}$ of $w$ codes $W$. Then, during inversion, at each step, we obtain the $k$ nearest neighbors of the current latent code estimate $kNN(w_i)$ from $\mathcal{S}_{\mathcal{W}}$, and use a softmax weighted average of the distance to nearest neighbors as a loss:

$$\mathcal{L}_{kNN} = \sum_{w_i^k \in kNN(w_i)} \|w_i - w_i^k\|_2 \cdot \frac{e^{-0.5 \frac{\|w_i - w_i^k\|_2}{\bar{d}_i^k}}}{D_i}, \tag{8}$$

where $D_i$ is the normalizing factor, and $\bar{d}_i^k$ is the mean distance to nearest neighbors:

$$D_i = \sum_{w_i^k \in kNN(w_i)} e^{-0.5 \frac{\|w_i - w_i^k\|_2}{\bar{d}_i^k}}, \quad \bar{d}_i^k = \sum_{w_i^k \in kNN(w_i)} \|w_i - w_i^k\|_2. \tag{9}$$

Figure 4 provides an intuitive visualization of our kNN loss in 2D. Our loss landscape does not penalize as much the exploration of the distribution and its boundaries. Further, it respects the non-isometric behavior of the $\mathcal{W}-$space. The soft-max weight ensures spatial continuity and a higher contribution for neighbors closer to the optimized latent (Eq. 8). To normalize the weights, we use the mean distance to the nearest neighbors ($\bar{d}_i^k$, Eq. 9). We provide the results of our optimization scheme and ablation studies demonstrating the benefits of our kNN loss in Sec. 5. Note that we use $k = 50$ for all our experiments.

# 4 Adapting Existing Inversion Methods

In this section we show how existing initialization and generator fine-tuning mechanisms can be adapted to JoIN to improve the inversion result and bridge the Sim-to-Real gap.

## 4.1 Encoder-guided Initialization

While our optimization-based approach produces faithful decompositions on synthetic data, the optimization can get stuck in local minima if the initial latent codes lie far from the target codes. To make our approach

more robust, we propose to use an encoder-guided initialization inspired by hybrid approaches of GAN inversion (Chai et al., 2021b; Bau et al., 2019c; Wei et al., 2021). For each component, we can independently train a pSp network (Richardson et al., 2021a) to encode a natural image into a latent code $w$ used by the pre-trained generator to recover the given component. At test time, given a natural image, the latent code produced by the encoder can be used as an initialization for the optimization.

### 4.2 Generator Fine-tuning using PTI

Since our component-wise GAN models are trained using only synthetic data, they are unable to generate the features corresponding to the real images. Thus, our method poses the challenge of Sim-to-Real gap, *i.e.* various features of the real images are not captured faithfully in the decomposition. As a remedy, we propose to carefully fine-tune the generators in such a way that they can generate decompositions of real images while still maintaining their original priors. Indeed, approaches leveraging generator fine-tuning have become popular (Roich et al., 2021; Feng et al., 2022) where first a $w-$code is estimated using an off-the-shelf inversion technique, and then with the fixed $w$ pivot, the generator is updated to obtain close to zero reconstruction errors. Such approach is also known as Pivotal Tuning for Inversion (PTI) in the literature.

In our case, the generator fine-tuning (PTI) can be defined as:

$$\underset{\theta_{G_i}, i \in 1 \dots n}{\arg\min} \mathcal{L}\left(I^*, f\left(T_1^{-1}\left(G_1\left(\widehat{w_1}\right)\right), \dots, T_n^{-1}\left(G_n\left(\widehat{w_n}\right)\right)\right)\right), \tag{10}$$

where $\theta_{G_i}$ represents trainable parameters of the generator $G_i$ that corresponds to the component $C_i$, and $\widehat{w_i}$s are output of the optimization.

While direct fine-tuning allows the generator to overfit to the target image perfectly, it distorts the GAN manifold. In our case, allowing multiple GANs to fine-tune at once can result in loss of the priors and much greater cross-contamination (see Fig. 11b *w/o D Loss*) since we do not have any component-wise supervision for individual generators. Here, we leverage interesting fact about the presence of strong priors in GAN: since the discriminator of the GAN is trained to discriminate between real and fake samples, its knowledge about the *realness* of a sample can be employed to ensure the prior remains preserved during generator fine-tuning. In practice, one can use the discriminator loss to regularize the fine-tuning (Feng et al., 2022). While fine-tuning the generator around the $\widehat{w_i}$ pivot, we apply a new localized discriminator loss $\mathcal{L}_{LD}$, we refer to it as *D Loss*.

Specifically, we sample $n_a$ number of anchor latent codes in the close vicinity of our fixed $w-$codes by interpolating $\widehat{w_i}$ with randomly sampled codes $w_k$. At each fine-tuning step, we pass the anchor codes through the generator and compute D loss on the resulting images, keeping the discriminator frozen:

$$\mathcal{L}_{LD} = \sum_i \sum_{k=1}^{n_a} D_i(G_i((1-\beta)\widehat{w_i} + \beta w_k)), \tag{11}$$

where $w_k$s are random latent codes, $G_i$ and $D_i$ are the generator and the discriminator for the component $C_i$ and $\beta$ determines the extent of interpolation, controlling the localization of our loss. Our fine-tuning formulation thus becomes:

$$\underset{\theta_{G_i}, i \in 1 \dots n}{\arg\min} \mathcal{L}_G + \lambda_{LD}\mathcal{L}_{LD}, \tag{12}$$

where $\mathcal{L}_G$ is the generator loss as defined in Eq. 10. As discussed in the ablation study (Sec. 5.4), the $\mathcal{L}_{LD}$ loss ensures that images with latents close to the pivot maintain realism, preserving the generator's priors during the optimization even as multiple generators are fine-tuned together.

## 5 Experiments

In this section, we present a set of experiments that showcase the efficacy of our method. We provide intrinsic image decomposition results on two different datasets: materials, and faces. We also provide results on the

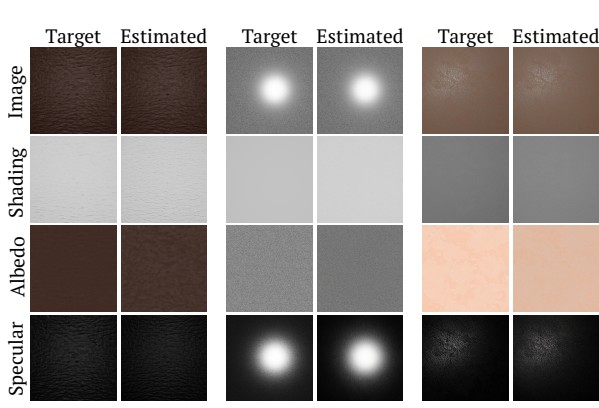

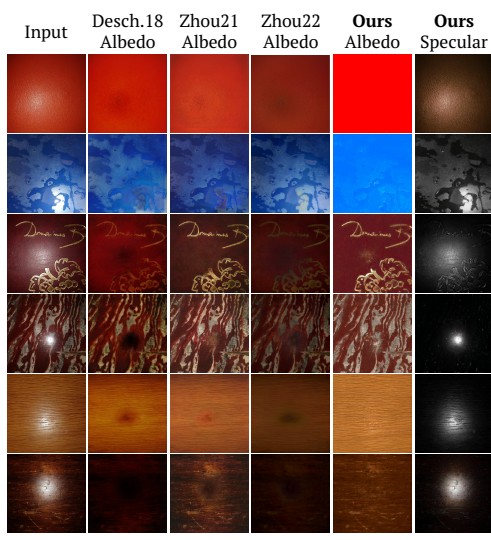

(a) **Decomposition on synthetic materials.**   (b) **Comparisons on real materials.**

Figure 5: **(a)** We show three examples of decomposition on synthetic materials. For each case the first column, from top to bottom, contains target image (used for optimization), groundtruth shading & albedo while the second column contains our estimates. We are able to correctly separate the highlights and to discriminate between darker shading and darker albedo. **(b)** Our method generalizes to real materials as well – our GAN priors allow us to better separate the highlight from the albedo leading to fewer visual artifacts. Our recovered specular is shown on the right.

dataset of rendered shapes and indoor scenes in appendix. Note that for all our experiments, we use only synthetic data for training our GANs while showing results on both synthetic and real images along with comparisons with existing methods and ablation studies. We plan to release our code in future.

## 5.1 Preliminaries

For all our experiments we use StyleGAN2 (Karras et al., 2020b) to train the generators and a pSp (Richardson et al., 2021b) encoder for initialization. In appendix (Sec. B), we also provide additional decomposition experiments using StyleGAN-XL (Sauer et al., 2022) to depict generalizability of our model on complex image distribution of indoor scenes. All our generators and encoders are trained using their publicly available codebases and with the standard choices of the hyperparameters.

Our optimization-based inversion method remains the same for all the datasets with a learning rate of 0.1, kNN loss weight of 0.0001, and $k = 50$. We run the optimization for 1000 steps.

The decomposition results are evaluated using several metrics such as MSE, LPIPS, and PSNR for the synthetic images that provide ground truth. We also provide comparisons with several state-of-the-art methods that are used in the literature for intrinsic image decomposition. The results demonstrate the effectiveness of our proposed method in decomposing the input images into their intrinsic components and the ability to handle real images.

## 5.2 Experiments on Materials Data

The Materials dataset is composed of highly realistic renderings of synthetic material tiles as done in Deschaintre et al. (2018). We obtain the albedo map directly from the material and compose the shading and specular components from blender direct and indirect rendering passes. More details on this dataset are given in the appendix (Sec. C). For this dataset, we show results for decomposition into 3 components (albedo, shading, and specular; Eq. 6) and operate at the resolution of $512 \times 512$. For synthetic test images, we show results in Fig. 5a and quantitative comparisons to several recent SVBRDF estimation methods (Deschaintre et al., 2018; Zhou & Kalantari, 2021; Zhou & Khademi Kalantari, 2022) on the task of albedo recovery in

**Quantitative comparisons**: We obtain the reconstruction error for albedo estimation across the testset of 100 images for both the synthetic materials (in Tab. 1) and synthetic faces (in Tab. 2) datasets, and provide comparisons with existing decomposition approaches using various metrics.

Table 1: **Quantitative comparisons on synthetic materials testset**.

| On synthetic materials | Estimated Albedo | | |
|---|---|---|---|
| | MSE↓ | LPIPS↓ | PSNR↑ |
| Desch18 | 0.021 | 0.314 | 19.53 |
| Zhou21 | 0.020 | **0.293** | 20.92 |
| Zhou22 | 0.021 | 0.296 | 20.18 |
| **Ours** (Opt. + kNN loss) | **0.014** | 0.326 | **21.58** |

Table 2: **Quantitative comparisons on Lumos synthetic faces testset**

| On synthetic faces | Estimated Albedo | | |
|---|---|---|---|
| | LPIPS↓ | MSE↓ | PSNR↑ |
| Total Relighting | 0.2220 | 0.0644 | 17.1470 |
| PieNet | 0.1843 | 0.0501 | 19.4761 |
| InverseRenderNet++ | 0.3794 | 0.1550 | 14.5227 |
| NextFace | 0.2034 | 0.0791 | 19.0699 |
| Latent Intrinsics | 0.1939 | 0.1053 | 15.7492 |
| **Ours:** | | | |
| Optimization | 0.1311 | 0.0417 | 22.4763 |
| + kNN loss | 0.1047 | 0.0357 | 23.5331 |
| + Encoder | 0.0887 | 0.0345 | 24.1467 |
| + PTI w/o D Loss | 0.0784 | 0.0155 | 24.1160 |
| + PTI w/ D Loss | **0.0765** | **0.0072** | **27.3401** |

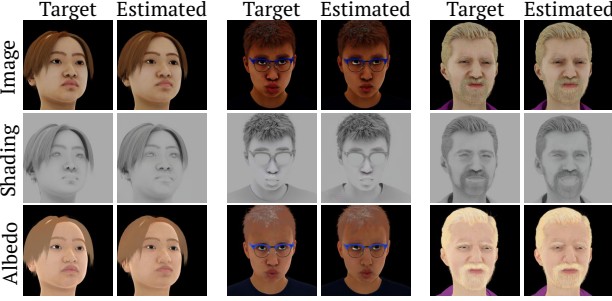 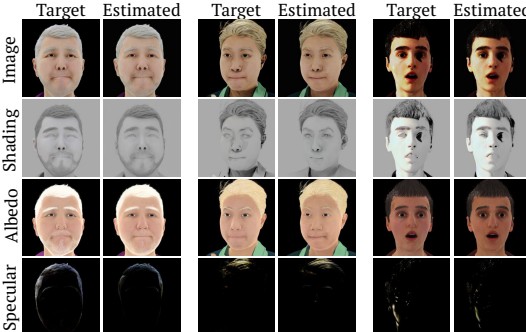

Figure 6: **Decomposition results for Lumos faces dataset.** On synthetic data, our estimated components are close to the ground truth, generator tuning is not needed when testing data is close to the GAN training distribution. **Left:** Results of our method for decomposing the image into two components: albedo and shading. **Right:** Using three components allows to recover subtle specular effects.

Tab. 1. We also show competitive results on albedo map recovery both quantitatively and qualitatively on real images, taken from (Deschaintre et al., 2019), visible in Fig. 5b. Our GAN priors allow to correctly separate flash highlights from the diffuse component and to recover plausible content under the highlights in the albedo map while correcting for the lighting fall-off. Additional results are in appendix (Sec. E).

## 5.3 Experiments on Faces Data

Faces are notoriously adapted to the usage of GANs and to cater to such a use case, we use the synthetic Lumos dataset (Yeh et al., 2022) to train our GANs, we defer more discussions on the dataset to the appendix.

First, we show results on test images from the dataset in Fig. 6(left) and Fig. 6(right) where we use respectively two and three components. For this experiment, we use the encoder initialization and our optimization without generator fine-tuning (PTI). As can be seen in these figures our method is able to recover accurate albedo, shading, and specular maps and does not suffer from cross-channel contamination.

As the synthetic dataset does not reproduce the complexity and diversity of real faces, for real images, we add the generator fine-tuning (w/ local D loss) step presented in Sec. 4.2. We show results on real faces taken from the FFHQ dataset in Fig. 7a and compare our method to five recent relighting methods, Total Relighting (Pandey et al., 2021), Lumos (Yeh et al., 2022), InverseRenderNet++ (Yu & Smith, 2021), PieNet (Das et al., 2022a), Nextface (Dib et al., 2021a;b; 2022), and Latent Intrinsics (Zhang et al., 2024) that seek to recover face albedo as part of their relighting pipeline, as shown in Fig. 7b. In all the cases, the predicted components by our method maintain the priors of each GAN. We can observe that our method works better at producing shading-free albedo images that exhibit fewer shading variations around the neck, nose, nostrils, and mouth.

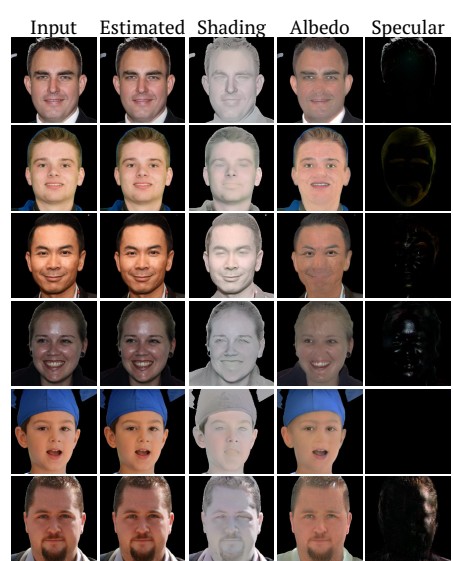

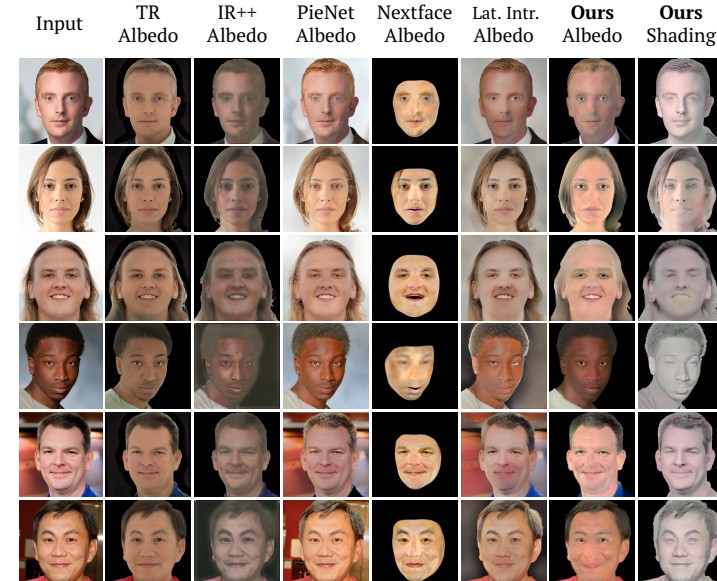

(a) **Decomposition results on real faces.**
(b) **Comparisons on real face images.**

Figure 7: **(a)** Using generator fine-tuning, our method generalizes to real faces and correctly separates albedo, shading, and specular effects while reconstructing the input correctly. **(b)** On real faces, our approach recovers more faithful albedo by removing shadows, and preserving the appropriate skin tones as compared to competing methods.

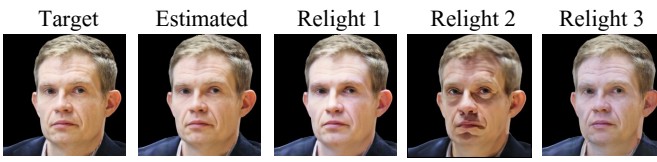

Figure 8: **Face relighting results.** Once the decomposition of a real face is obtained, we modify the shading component using GAN editing technique Shen & Zhou (2021) on the shading generator obtaining plausible relighting edits.

Table 3: **Effectiveness of kNN Loss in inverting a single GAN**

| On CelebA-HQ test set | Inversion on single GAN trained on FFHQ | | |
|---|---|---|---|
| | **LPIPS↓** | **MSE↓** | **PSNR↑** |
| Optimization | 0.384 | 0.095 | 16.908 |
| Opt. + in-domain loss | 0.384 | 0.094 | 16.990 |
| Opt. + kNN loss (ours) | **0.379** | **0.092** | **17.120** |

For quantitative comparisons, we use a testset of synthetic images from Lumos dataset, and compare reconstruction errors for albedo recovery for all variants of our approach with four competing methods in Tab. 2. We see that our method achieves the lowest errors. Additional results are provided in appendix (Sec. E).

**Application to relighting.** GANs are known for their rich image manipulation capabilities. Our modular approach allows leveraging such capabilities to modify each image component separately using GAN's latent space editing, making it suitable for image modifications such as relighting. We show real image relighting results in Fig. 8. To generate these results, after obtaining the inversions for albedo, shading, and specular, we use an off-the-shelf latent space editing method SeFa (Shen & Zhou, 2021) on the shading generator to modify the shading image while keeping the other two components fixed. StyleGAN's disentangled representation helps with identity preservation. As seen in Fig. 8, our method produces varied novel lighting.

## 5.4 Ablation Studies

**Advantage of independently trained GANs.** Training GANs independently makes our pipeline modular and flexible: different components can be trained with different data sources, and new components can be easily introduced in the forward model without re-training the entire pipeline. We showcase such use case by adding the specular component to the albedo-shading forward model (Fig. 6, 7a, 5a, 5b).

We also show that it is easier to train and invert GANs independently, further preventing cross-contamination during inversion. To demonstrate this, we train a single multi-component GAN that learns to generate both

Table 4: **Quantitative comparisons** for different variations of our approach on synthetic Faces.

| | On synthetic Faces (Lumos dataset) | | | | | | | | | | | |
| --- | --- | --- | --- | --- | --- | --- | --- | --- | --- | --- | --- | --- |
| | Albedo | | | Shading | | | Specular | | | Image | | |
| | LPIPS↓ | MSE↓ | PSNR↑ | LPIPS↓ | MSE↓ | PSNR↑ | LPIPS↓ | MSE↓ | PSNR↑ | LPIPS↓ | MSE↓ | PSNR↑ |
| Inversion w/ single GAN for both albedo and shading | 0.1954 | 0.1269 | 14.166 | 0.2657 | 0.07021 | 18.245 | - | - | - | 0.09342 | 0.03133 | 24.466 |
| Optimization | 0.1311 | 0.0417 | 22.476 | 0.2832 | 0.0518 | 18.863 | 0.2544 | 0.0336 | 32.320 | 0.0879 | 0.0308 | 24.296 |
| Opt. + in-domain loss | 0.1209 | 0.0400 | 22.589 | 0.2738 | 0.0497 | 19.159 | 0.2444 | 0.0332 | 32.833 | 0.0876 | 0.0302 | 24.164 |
| Opt. + kNN loss | 0.1047 | 0.0357 | 23.533 | 0.2603 | 0.0487 | 20.911 | 0.2361 | 0.0320 | 32.905 | 0.0868 | 0.0301 | 24.114 |
| Encoder + Opt. + kNN loss | 0.0887 | 0.0345 | 24.146 | 0.2039 | 0.0414 | 21.032 | 0.1722 | 0.0282 | 33.409 | 0.0713 | 0.0306 | 26.445 |
| Encoder + Opt. + kNN loss + PTI w/o D Loss | 0.0784 | 0.0155 | 24.116 | 0.2775 | 0.0519 | 18.165 | 0.1550 | 0.0278 | 32.838 | **0.0136** | **0.0011** | **33.545** |
| Encoder + Opt. + kNN loss + PTI w/ D Loss | **0.0765** | **0.0072** | **27.340** | **0.1703** | **0.0285** | **22.475** | **0.0774** | **0.0251** | **34.547** | 0.0194 | 0.0022 | 32.517 |

albedo and shading images simultaneously by producing a 6-channel output: 3 channels each for albedo and shading. We increase the number of trainable parameters by 2× for fair comparison with our method.

First, as shown in Fig. 9a, such joint training leads to inferior image generation compared to independently-trained albedo and shading GANs. With a single GAN, albedo and shading are leaking into each other, this is also measured by higher FID scores. Second, having the same network generating both albedo and shading means that the information is entangled up to the last layer, leading to cross-contamination during inversion as shown in Fig. 9b. Our observation is also supported by the quantitative results provided in Tab. 4.

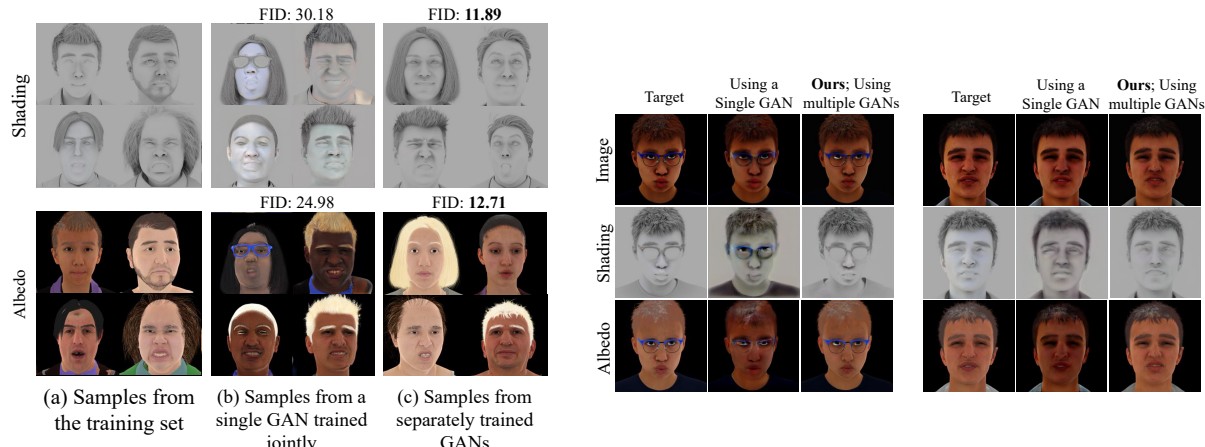

(a) **Single GAN vs separated GANs as prior.**  (b) **GAN inversion on single vs separated GAN priors.**

Figure 9: **(a)** Training separate GAN for each component results in better generation quality as opposed to training a single GAN for learning both components. **(b)** Inversion on a single GAN trained to generate both albedo and shading leads to cross-contamination issues (note the color leakage in shading and shadow effects in albedo). We mitigate this issue by training independent GANs for each component.

**Comparisons with DoubleDIP.** We provide qualitative comparisons with doubleDIP (Gandelsman et al., 2019) in Fig. 10a. As shown, untrained priors-based doubleDIP results in severe cross-contamination and loss of details on both the synthetic and real datasets as the underlying assumption of dissimilarity across the components does not hold. Our method produce accurate decompositions in both the cases.

**Effectiveness of kNN loss.** To evaluate the effectiveness of our newly proposed kNN-loss, we first provide experiments for the more general single GAN inversion setup: we use a StyleGAN2 model trained on natural face images from FFHQ dataset (Karras et al., 2019a), and invert CelebA-HQ testset (Karras et al., 2018) images using optimization-based inversion (Karras et al., 2020b). As shown in Tab. 3, using kNN loss as a regularization improves the performance over direct optimization and optimization with in-domain loss. Further, we verify the effectiveness of kNN loss for IID task using joint inversion on synthetic faces. As shown in Fig. 10b, kNN loss prevents cross contamination which is prevalent when using in-domain loss

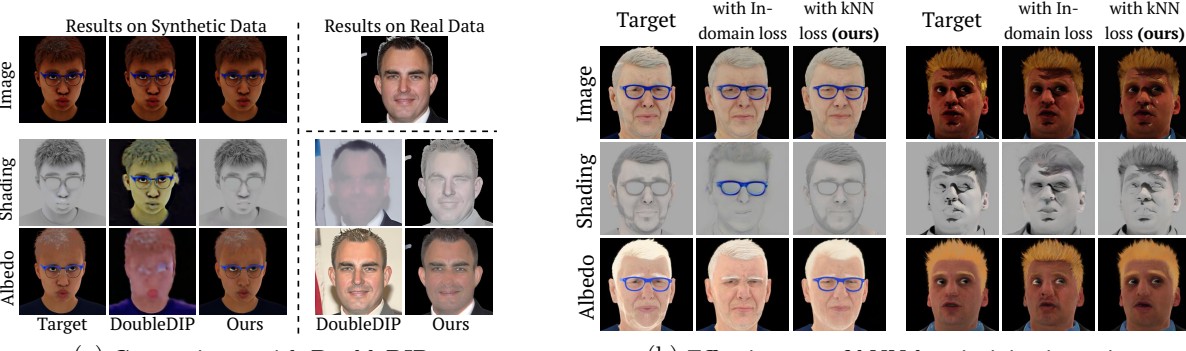

(a) **Comparisons with DoubleDIP.**    (b) **Effectiveness of kNN loss in joint inversion.**

Figure 10: Ablation Results for (a) comparisons with DoubleDIP, and (b) effectiveness of kNN loss in joint inversion.

(notice glasses and shadows appearing in albedo), and improves the detail preservation (note mouth, beard, and hairstyle). Such improvements are supported by superior quantitative performance in Tab. 4.

**Full method ablations.** Finally, we present a comprehensive set of ablations for our joint inversion of multiple GANs. We provide qualitative results (Fig. 11a) and quantitative evaluations for intrinsic image decomposition on the Lumos Faces dataset in Tab. 4, second part. We first evaluate our method using only direct Optimization (Opt.). We then add the in-domain loss discussed in Sec. 3.4 and then replace it with our kNN loss. As can be seen in Tab. 4 the kNN Loss allows better recovery of the intrinsic components by maintaining the latent codes within domain, it also outperforms the in-domain loss.

We then add the encoder initialization from Sec. 4.1 to the pipeline and finally add the generator fine-tuning (PTI), without, then with, the D Loss as discussed in Sec. 4.2. Our evaluation shows that the encoder initialization brings a clear improvement in both reconstruction quality and component estimations. The PTI without D Loss does improve the reconstruction of the input image but performs poorly in component estimation, whereas, with the D Loss, the PTI provides the best results for the decomposition while still improving the image reconstruction quality. A qualitative visualization of this difference is visible in Fig. 11b.

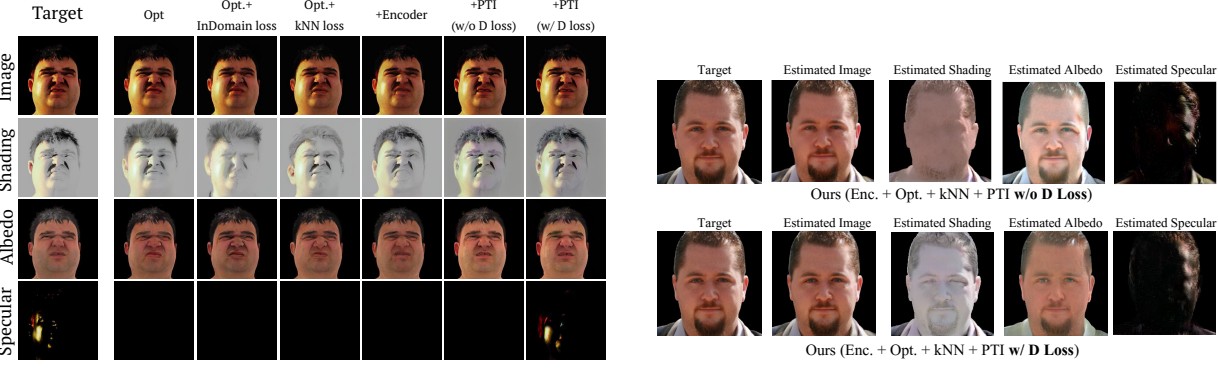

(a) **Full method ablations on synthetic faces.**    (b) **Ablation on importance of local D loss.**

Figure 11: **(a)** Our decomposition improves as we (from left to right) replace in-domain loss with kNN loss; initialize with and encoder; fine-tune generators using PTI; and use local D loss during the PTI fine-tuning step. **(b)** Local D loss preserves the generator priors by preventing the local manifold from distortions. Above we show IID on a real face image without (top) and with (bottom) local discriminator loss.

## 5.5 Storage, Memory, and Computational Costs

**Runtime.** Our approach adapts efficient GAN inversion techniques ($w$−optimization, encoder-based initialization, and Pivotal Tuning Inversion (PTI)) for joint GAN inversion, with our novel kNN and local discriminator losses adding minimal overhead ($\approx 15\%$ and $\approx 10\%$ to the optimization and PTI runtimes,

respectively). This is achieved through GPU-accelerated faiss library (Johnson et al., 2019) for nearest neighbor search, which takes $< 2$ms to compute nearest neighbors each optimization step (for $k = 50$). On a single NVIDIA A40, JoIN takes 190 seconds for inference on a single image (84s for optimization, 1s for encoder, 105s for PTI). While slower than feed-forward methods like InverseRenderNet++ (2s) and PieNet (1s), JoIN is considerably faster than optimization-based methods like NextFace (320s), and multiple priors methods such as DoubleDIP (878s), UIDNet (950s) and diffusion-based DMPlug (635s). Crucially, JoIN is fully amortizable, enabling concurrent processing of 8 images on a single NVIDIA A40, reducing per-image inference time to 24 seconds (11s for optimization, $< 1$s for encoder, 13s for PTI).

**Storage and Memory.** JoIN requires $\approx 800$MB for model storage, lower than PieNet (2200MB), NextFace (1300MB), and InverseRenderNet++ (1200MB). GPU memory usage during inference for our method is 4GB, which is on par with other competing methods.

In summary, JoIN strikes a favorable balance between accuracy and efficiency. While feed-forward methods like InverseRenderNet++ and PieNet offer faster inference, they lack the flexibility to adapt to different forward models without retraining, a key advantage of our method. Compared to optimization-based methods like NextFace and prior-based methods like UIDNet, JoIN achieves significant runtime reductions, particularly considering the amortization.

## 6 Limitations and Future Work

In this paper, we introduced JoIN – a new method to solve inverse problem of image decomposition using a bank of GANs as priors. We show that it is possible to retain GANs distribution priors while jointly inverting several GANs at once. Nonetheless, our method inherits some drawbacks of GAN inversion methods. First, it is slower than feed-forward methods with runtimes of a few seconds (detailed discussion is in appendix Sec. 5.5). Second, currently, training GANs on unaligned distributions remains a complex task and our method can be limited by the GAN training quality, hence it is more adapted to distributions that GANs can model well such as face images (we demonstrate generalizability of our method to newer GAN models and complex indoor scenes in appendix Sec. B). Our framework of using bank of GANs as a prior is not limited to only the problem of IID, and can be extended to similar inverse problems in other domains of engineering such as audio source separation as a part of future work. An interesting avenue for future work is exploring the replacement of GAN priors with diffusion models, which have shown impressive results in various image generation and inverse problems, to investigate their potential for improving the accuracy and robustness of intrinsic decomposition. Particularly for larger and more diverse datasets, ability of diffusion models to capture complex data distributions may prove advantageous. As new techniques emerge, we believe our method will be applicable to broader datasets and could be tested on a wider variety of problems.

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

## Appendix

We provide following as appendix to support the discussion and results of the main paper:

- In Sec. A, we discuss the intrinsic image decomposition **experiments on primeshapes dataset**. We come up with such fully synthetic dataset of simple shapes with variations of lights and materials in order to demonstrate the efficacy of our method with the two-component model (albedo and shading). The qualitative results are shown in Fig. 12.

- In Sec. B, we discuss the **generalizability of our approach to newer GAN models** and more complex data distributions such as indoor scenes. We also provide qualitative and quantitative results for Hypersim dataset of indoor scenes using StyleGAN-XL model (see Fig. 13 and Tab. 5).

- In Sec. C, we discuss our **pipeline for generating and processing our datasets**. Specifically, we discuss how we generate Primeshapes and Materials synthetic datasets using blender in python, and pre-processing steps involved for Lumos synthetic face dataset. We also discuss the **responsible usage of the human data**.

- In Sec. D, we discuss the responsible use of human data and other ethical considerations pertaining to our work.

- In Sec. E we provide **additional results** for experiments discussed in the main paper to present supporting evidence for the claims made. In particular, we provide:
  - Additional qualitative results for synthetic and real materials in Fig. 15, 16.
  - Additional qualitative results for synthetic and real faces in Fig. 17, 18, 19.
  - Additional qualitative comparisons with competing methods on real faces in Fig. 20.
  - Additional results for image relighting on faces using our method in Fig. 21.
  - Additional qualitative results for synthetic indoor scenes on hypersim testset using StyleGAN-XL in Fig. 22, along with qualitative comparisons with baselines in Fig. 23.

## A  Experiments on Primeshapes data

We start with the simpler case of decomposing images for the Primeshapes dataset a fully synthetic dataset that contains rendered images of a variety of primitive shape objects under varying lighting conditions. We use this dataset to demonstrate the efficacy of our method with the two-component model (albedo and shading). We produce our decomposition using only optimization. We initialize the latent codes $w_i$s with the mean latent code $\bar{w}_i$ of each GAN and apply the kNN loss in order to keep the $w-$code estimates in domain. As can be seen in Fig. 12, our method is able to achieve successful decomposition for this dataset retaining each GANs priors and without cross-contamination. Note that while we show the target albedo and shading for each example, only the target image is used during optimization.

## B  Generalizability to Newer GAN Models and Results on Indoor Images

The key idea of our method is GAN inversion, which is widely supported by most of the GAN models. While we chose StyleGAN2 to demonstrate the efficacy of our method owing to its superior performance and stable training, our method is generalizable to newer and more powerful GAN variants such as StyleGAN-XL Sauer et al. (2022). With powerful GAN variants, we are also able to model more complex image distributions such as indoor scenes with multiple objects. Here we show experiments on Hypersim dataset Roberts et al. (2021) of indoor scenes using StyleGAN-XL that supports complex image distributions. Hhypersim dataset provides photorealistic synthetic images of complex indoor scenes along with their rendered albedo and shading. Similar to the experiments on faces, we use optimization with kNN loss for decomposing synthetic scenes, and add generator fine-tuning for real scenes. In Fig. 13, we show the faithful decomposition results obtained using our method on hypersim dataset using StyleGAN-XL model. We also provide quantitative comparisons with existing methods for albedo reconstruction in Tab. 5 indicating significant performance improvements. Additional results are provided in Sec. E.

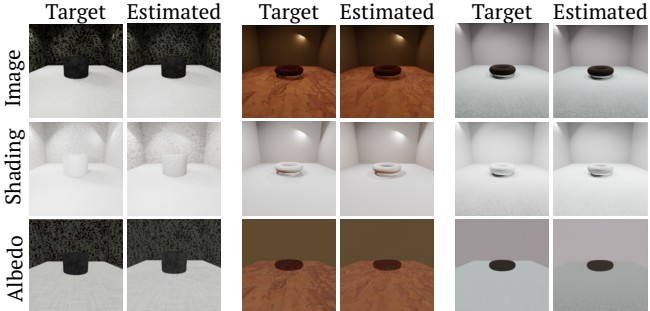

Figure 12: **Decomposition results for Primeshapes dataset.** Three examples are shown. For each, we show on the left column, from top to bottom, the target Image used for optimization, ground truth shading, and albedo for reference. The right column contains our estimates.

## C    Details of Rendered Datasets

**Data preparation pipeline.** We use the blender python package Blender Online Community (2023) to generate both the Primeshapes and Materials synthetic datasets. We make use of rendering and passes functionalities in blender python to render 11 components for each scene. Fig. 14 shows the relationship between these 11 components, and how they can be combined to obtain the image. We keep the pre-processing steps the same for both the Primeshapes and Materials datasets: we use the diffuse color component as albedo; we add diffuse direct and diffuse indirect to obtain shading; and for specular, we multiply glossy color with the sum of glossy direct and glossy indirect. We ignore the transmission, emission, and environment components in our inversion as they do not contribute significantly to the final image in our setting. Thus our composed image, used as the target for inversion, is obtained with:

$$\text{final image} = \text{shading} \cdot \text{albedo} + \text{specular} \tag{13}$$

**Inverse color-transform operation (tone-mapping).** Since the renderings generated using blender are in linear space, we convert them in the color space by applying suitable inverse color-transform operation (more widely known as tone-mapping in the literature) to each of the image components in order to use them in the GAN/encoder training. Our choice of tone-mapping functions is standard, and the same for both Primeshapes and Materials datasets. For albedo, we use gamma/sRGB tone-mapping with $\gamma = 2.4$, while for shading and specular, we use Reinhard global tone-mapping Reinhard et al. (2002) as these signals are inherently high-dynamic range.

Table 5: **Quantitative comparisons on Hypersim dataset of indoor scenes using StyleGAN-XL**

| On Hypersim synthetic images | Estimated Albedo | | |
|---|---|---|---|
| | LPIPS↓ | MSE↓ | PSNR↑ |
| PieNet | 0.312 | 0.233 | 12.907 |
| InverseRenderNet++ | 0.453 | 0.314 | 12.217 |
| **Ours** | **0.184** | **0.0203** | **23.411** |

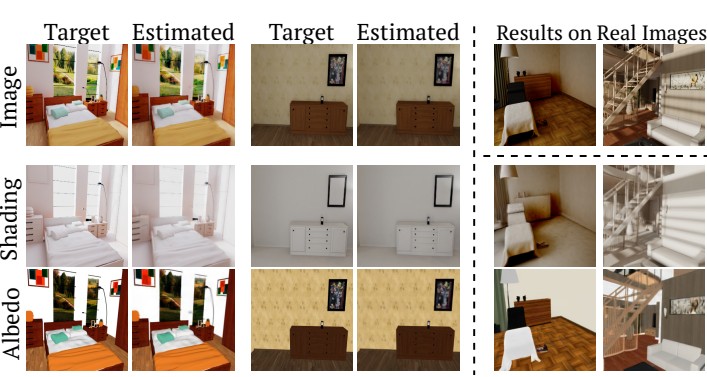

Figure 13: **Decomposition results on synthetic and real indoor scenes** using our method on StyleGAN-XL model Sauer et al. (2022) trained with hypersim dataset Roberts et al. (2021). Here we show decomposition on synthetic images from hypersim testset (left) and on unseen real images (right) to demonstrate the generalizability of our approach.

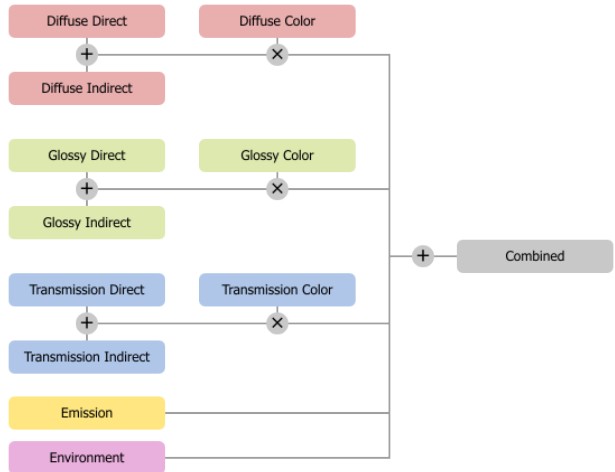

Figure 14: **Components available in our dataset.** (Image source: Blender documentation Blender Online Community (2023)).

## C.1 Primeshapes Synthetic Dataset

The Primeshapes dataset is a fully synthetic dataset containing rendered images of a variety of primitive shapes such as spheres, cones, cubes, cylinders, and toroids placed in a room with textured walls and lit by spotlights. Random materials chosen from a large collection of materials are assigned to the objects and the walls. The location of the lights is selected at random while the camera angle is kept fixed for all the images allowing easier training of GANs. This dataset contains $100,000$ rendered images at the resolution of $256 \times 256$, out of which $70,000$ images are used to train the StyleGAN generators and pSp encoders for individual components by using diffuse color as albedo, and the sum of direct and indirect shading as shading. The remaining $30,000$ images are used as unseen testing images. To keep the problem simple in the context of the Primeshapes dataset, the inversion is run on a target image that does not use the glossy layers, thus using only the two components we aim at recovering (albedo and shading).

## C.2 Materials Synthetic Dataset

The Materials dataset is a fully synthetic dataset containing $80,000$ renderings of materials taken from the Substance 3D collection. These synthetic materials are rendered with simulated flash lighting. Out of the full dataset, $1,000$ images are left out to serve as a test set. To generate each sample, we randomly pick a distance to the material, an orientation of the surface, and a random light position close to the camera to simulate camera-light misalignment. We use a sphere light source with varying diameters and intensities. Images are rendered at $512 \times 512$.

## C.3 Lumos Synthetic Faces Dataset

The Lumos dataset (Yeh et al., 2022) provides thousands of synthetic face images captured in virtual light-room created inside a 3D design software. For each image, various intrinsic sub-components such as albedo, shadow, sub-surface scattering, specular etc. are also provided. We combine these sub-components to obtain albedo, shading, and specular maps used to train our GANs. We leverage the facial image alignment method used for the FFHQ dataset (Karras et al., 2018; 2019a) to pre-process each image in the Lumos dataset. Such pre-processing ensures that all the facial images are aligned in the same fashion, thus improving GAN training and allowing us to test our method on the real images from FFHQ.

The Lumos synthetic dataset provides several image components for each face such as shadow, sub-surface scattering, coat, and sheen. While we use the albedo component directly from the dataset, we use the

following strategy to obtain the shading and specular components for each image and train our GANs:

$$\text{shading} = \text{shadow} + \frac{\text{(sub-surface scattering)}}{\text{albedo}} \tag{14}$$

$$\text{specular} = \text{specular} + \text{sheen} + \text{coat} \tag{15}$$

We continue to use the Eq. 13 for faces as well in order to combine albedo, shading, and specular to get the final image. Our tone-mapping functions for the Lumos dataset are similar to the Materials dataset: for albedo we use gamma/sRGB tone-mapping with $\gamma = 2.4$, while for shading and specular, we use Reinhard tone-mapping (Reinhard et al., 2002).

## D    Broader Impact Concerns

### D.1    Responsible Use of Human Data

We perform our experiments on images from four different domains: primeshapes, materials, faces, and indoor scenes. For each case, we use fully synthetic datasets for training the GAN models on individual intrinsic components, and use synthetic and/or real data for testing and evaluation of our model. None of the synthetic datasets we use contain any human subject data or personally identifiable information. The only scenario where we potentially use human subject data is for evaluation of our method on real faces using FFHQ dataset.

In particular, for faces, we use Lumos synthetic face dataset for training, and a small number of real images from FFHQ dataset Karras et al. (2018; 2019a) for evaluating our model. FFHQ dataset contains $70,000$ images of human faces collected from an online image sharing platform named Flickr. This dataset is diverse, featuring people of various ages, ethnicity, and backgrounds. It even includes a wide range of accessories like glasses, sunglasses, and hats. The source for these images was Flickr, so it's important to be aware of potential biases present on that platform that maybe inherited by the dataset. During the collection of the dataset, it was made sure that only those images with permissive licenses are included. Finally, to ensure the dataset's quality, automated filters and human reviewers (via Amazon Mechanical Turk) removed any irrelevant images like statues or pictures of pictures. It is important to note that this dataset is not intended to be used for any facial recognition technology. Licensing and usage information about the dataset is available on its webpage[1]. We are committed to the responsible usage of human subjects data, and made sure we follow all the licensing requirements and policies in our experiments with FFHQ dataset.

### D.2    Ethical Considerations

Although our method focuses on intrinsic image decomposition, as discussed in Sec. 5, one can potentially combine our approach with GAN's latent space image editing to achieve image relighting on human faces. While such relighting does not change the underlying appearance of the face, potential broader impact concerns still exist. Realistic relighting could be misused to misrepresent the context of an image by subtly altering the perceived environment, potentially misleading viewers. Furthermore, even without altering facial features, our method might inadvertently reinforce biases present in the training data regarding lighting conditions and their association with certain demographics. Finally, relighting could complicate the authentication of images, making it more challenging to detect subtle forms of manipulation. Therefore, we advocate for responsible development and deployment of relighting technology, alongside ongoing research into detection methods, to mitigate these risks and maintain trust in visual media. Transparency about our method's limitations is crucial for fostering informed discussion and responsible use.

## E    Additional Results

We provide additional results and comparisons to support our claims, discussed in the main paper.

---

[1]https://github.com/NVlabs/ffhq-dataset

**Qualitative results on materials.** We provide additional results for decomposition on synthetic materials images in Fig. 15. As seen in the main paper, our model is able to recover albedo, shading, and specular components faithfully on synthetic images from Materials dataset. Further, we also provide additional comparisons with existing approaches on real material images in Fig. 16. Results are consistent with the qualitative and quantitative evaluations provided in the main manuscript.

**Qualitative results on faces.** We provide additional results for decomposition into 2 and 3 components on synthetic face images in Fig. 17 and Fig. 18 respectively. As seen in the main paper, our model is able to recover albedo, shading, and specular components faithfully on synthetic images from the Lumos dataset. Further, we also provide additional results on the real images from the FFHQ dataset in Fig. 19, along with the comparisons with other methods in Fig. 20. As discussed in the main paper, our method is able to preserve the GAN priors and separate the image into plausible components, without copying shading into the albedo image.

**Image relighting on faces.** As discussed in the main paper, our modular approach allows leveraging latent space editing capabilities of GANs to modify each image component separately, making it suitable for image modification such as relighting applications. We show additional image relighting results on real faces in Fig. 21. After obtaining the inversions for albedo, shading, and specular, we use an off-the-shelf latent space editing method SeFa (Shen & Zhou, 2021) on the shading generator to modify the shading image while keeping the other two components constant. StyleGAN's disentangled representation allows the identity of the person to remain relatively constant as we modify the shading image. As seen in Fig. 21, our approach can produce varied novel lighting.

**Qualitative results on synthetic indoor scenes.** We provide additional results for decomposition of synthetic indoor scenes from hypersim dataset using our approach on newer StyleGAN-XL model Sauer et al. (2022) in Fig. 22. As seen in the main paper, our model is able to recover albedo and shading components closer to the groundtruth, indicating the generalization of our approach to newer GAN models and to complex datasets such as hypersim. Further, we provide qualitative comparisons with competing methods (PieNet Das et al. (2022a), InverseRenderNet++ Yu & Smith (2019)) for albedo recovery on synthetic images in Fig. 23. One can notice that the albedo images recovered from competing methods have significant distortions and presence of shading elements, while our approach produces cleaner output without any shading contamination. Corresponding quantitative comparisons are provided in the main paper in Tab. 5 confirms the improved performance of our approach.

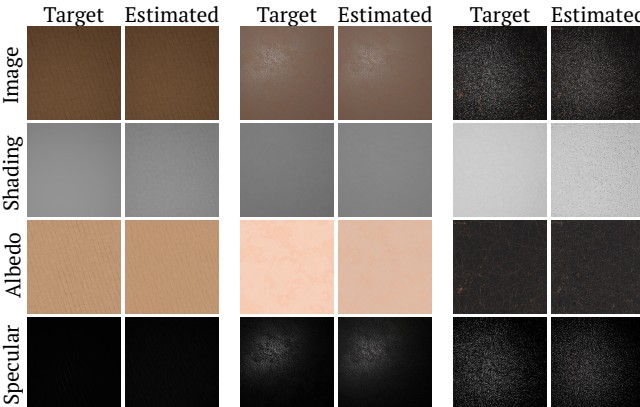

Figure 15: **Additional decomposition results for synthetic Materials.** Three examples are shown. Left column, from top to bottom, target Image (used for optimization), GT shading, and albedo. The right column contains our estimates. We are able to correctly separate the highlights and to discriminate between darker shading and darker albedo.

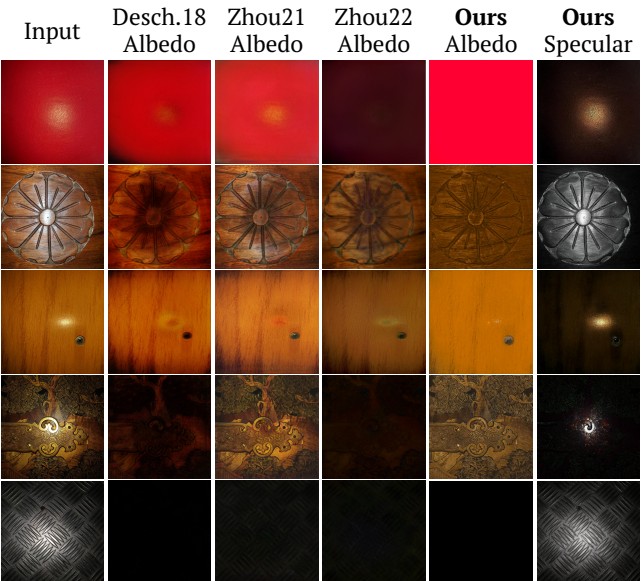

Figure 16: **Additional comparisons for Albedo estimation on real material pictures.** Our GAN priors allow us to better separate the highlight from the albedo leading to fewer visual artifacts. Our recovered specular layer is shown on the right for reference.

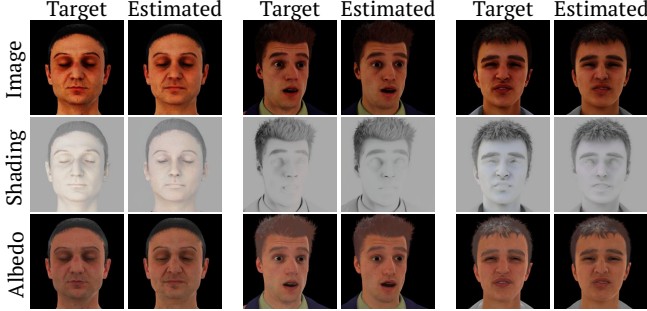

Figure 17: **Additional results for two components decomposition for Lumos Faces dataset.** On synthetic data, our component estimations are close to the ground truth, generator tuning is not needed when testing data is close to GAN training distribution.

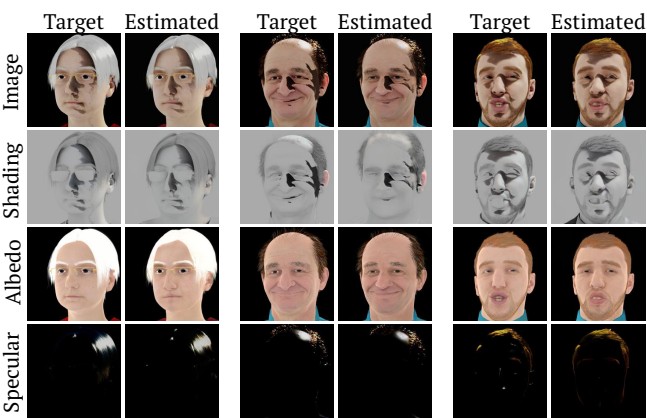

Figure 18: **Additional results for three components decomposition for Lumos Faces dataset.** Using three components allows to recover subtle specular effects.

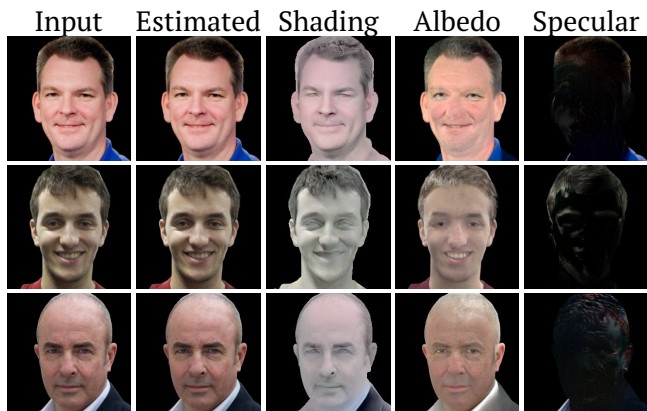

Figure 19: **Additional decomposition results for real faces.** Using generator fine-tuning our method generalizes to real faces and correctly separates albedo, shading, and specular effects while reconstructing the input correctly.

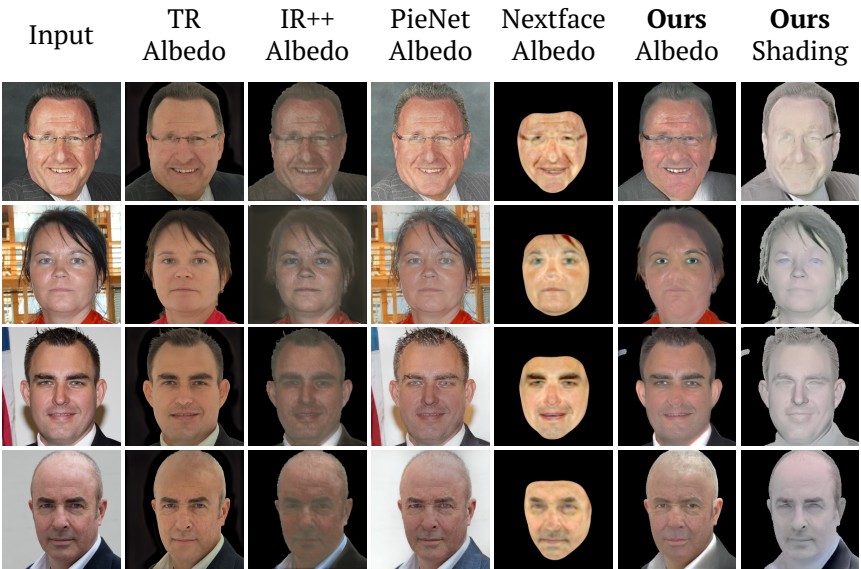

Figure 20: **Additional comparisons of our method on real face images.** On real faces, our approach can recover more faithful albedo by removing shadows, and preserving the appropriate skin tones as compared to other competing methods.

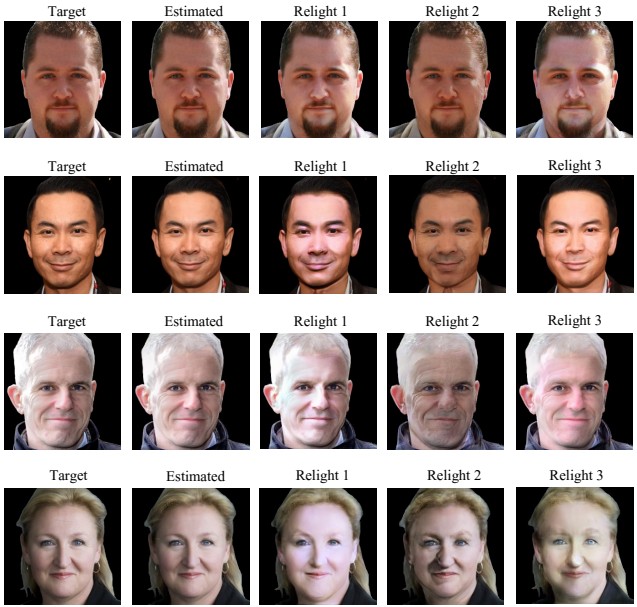

Figure 21: **Additional Face relighting results.** Once the IID is obtained, we modify the shading component using GAN editing technique Shen & Zhou (2021) on the shading generator obtaining plausible relighting edits.

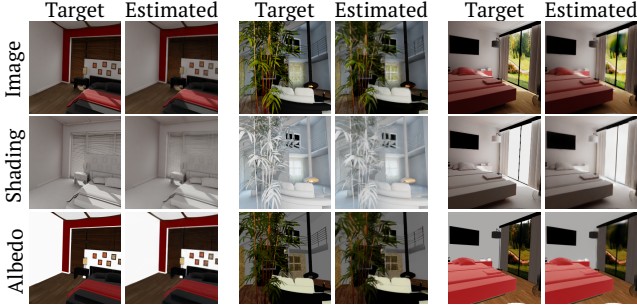

Figure 22: **Additional Decomposition results on synthetic indoor scenes** using our method on StyleGAN-XL Sauer et al. (2022) model trained with hypersim dataset Roberts et al. (2021). Note that our component estimations are close to the ground truth, demonstrating the generalizability of our approach to newer GAN models and complex datasets.

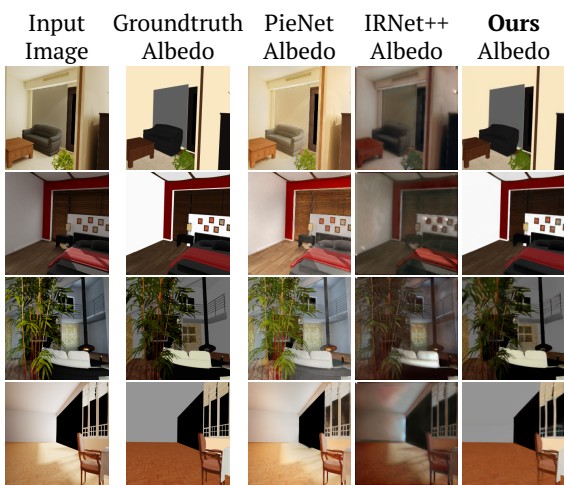

Figure 23: **Qualitative comparisons results for Albedo estimation on synthetic indoor scenes.** Notice that the albedo estimates produced by competing approaches have distortions and shading contamination, while our GAN priors allow us to better separate the highlight from the albedo leading to fewer visual artifacts.

