# OpenReview forum: "JoIN: Joint GANs Inversion for Intrinsic Image Decomposition"
_TMLR — Accepted by TMLR_

### Review · Reviewer_QPpm · 2024-11-11

**Summary Of Contributions:**

The authors introduce a method for intrinsic image decomposition that leverages multiple GANs as image priors. Each GAN is trained on a single image component (lighting, albedo, specularity, etc.) from synthetic data. These components are combined through a differentiable function into the final composite image. Then the authors use gradient descent (along with other standard GAN inversion techniques) to optimize the GAN latents so that the final composite image matches a target image that we wish to decompose. They also introduce a novel regularizer for W-space GAN inversion, K-nearest-neighbors loss, which outperforms another regularizer, in-domain loss, in their ablations. They also introduce a simple generalization of Pivotal Tuning for Inversion (PTI) which extends to their setting.

**Audience:**

Yes

**Claims And Evidence:**

Yes

**Requested Changes:**

N/A

**Strengths And Weaknesses:**

Paper is well written, easy to understand, proposes a novel technical solution to an important computer vision problem, with significant engineering effort applied to create and evaluate this solution. They obtain favorable results compared against other image decomposition methods. I have no criticisms or suggestions for improvement.

---

> ### Author Response · Authors · 2024-12-19
> **Authors' Response to Reviewer QPpm**
>
> We sincerely appreciate the reviewer's positive assessment of our work, particularly regarding the paper's clarity, the novelty of our approach using multiple GANs as priors, and the favorable results achieved. Thank you for the encouraging feedback and valuable time. We are happy to answer any further questions that may arise.

---

### Review · Reviewer_YG3X · 2024-11-19

**Summary Of Contributions:**

This paper proposes to use several independent GANs as prior to solve Intrinsic Image Decomposition (IID) problems, and introduces a kNN-based regularization term, which is a modified version of the in-domain loss, for joint GAN inversion. The proposed method can achieve good generalization on the real images by leveraging existing GAN inversion techniques such as generator fine-tuning.

**Audience:**

Yes

**Broader Impact Concerns:**

I have no concerns about the ethical implications of the work.

**Claims And Evidence:**

Yes

**Requested Changes:**

1. Provide comparisons between the proposed method with other competing methods for memory and computation costs.
2. Provide more literature review, including doubleDIP and its following works and diffusion inversion.

**Strengths And Weaknesses:**

Strengths:

* The paper writing is clear and easy to understand.
* The high-level idea, i.e., using different priors for different intrinsic components in IID, is valid.
* The empirical results are promising and ablation studies are sufficient.

Weaknesses:

* While the high-level concept makes sense, it has been extensively explored in the literature following the introduction of DoubleDIP [1] (over 300 following works). This idea has been applied to various inverse problems, including blind image deblurring [2] and phase retrieval [3]. Similarly, these papers use different image priors to reparameterize different components.

* There could be several problems of the proposed kNN loss:
1. Computational Overhead: Computing the k-nearest neighbors at each optimization step can be computationally expensive, especially in high-dimensional latent spaces or with large datasets.

2. Risk of convergence to suboptimal regions: Adhering too closely to local latent neighbors might limit global exploration and lead to suboptimal solutions. I guess that this paper chooses a very large $k$ (100k) to alleviate this issue, but if we choose a large $k$, then there would not be huge differences with the original in-domain loss.

3. Marginal improvements: As shown in Table 3, the benefits of the kNN loss over the original in-domain loss is very marginal, which is consistent with my argument above.

* I expect that the memory and computational costs of the proposed methods are much higher than previous methods because the proposed method needs to solve the optimization problems with multiple GANs and the kNN loss. This paper should at least conduction the comparisons between the proposed method with competitors in terms of both the memory and computational costs.

* GAN training is notorious for its instability and issues like mode collapse, whereas recent diffusion models effectively address these challenges and offer superior generation quality. This paper, however, does not justify the choice of GANs as a prior over diffusion models. Moreover, [4] has already introduced a GAN-inversion-style approach for solving inverse problems using pre-trained diffusion models. Specifically, Table 5 of [4] uses three different diffusion models for three different components for blind image deblurring with turbulence.


[1] Gandelsman, Y., Shocher, A. and Irani, M., 2019. " Double-DIP": unsupervised image decomposition via coupled deep-image-priors. In Proceedings of the IEEE/CVF conference on computer vision and pattern recognition (pp. 11026-11035).

[2] Ren, D., Zhang, K., Wang, Q., Hu, Q. and Zuo, W., 2020. Neural blind deconvolution using deep priors. In Proceedings of the IEEE/CVF conference on computer vision and pattern recognition (pp. 3341-3350).

[3] Zhuang, Z., Yang, D., Hofmann, F., Barmherzig, D. and Sun, J., 2022. Practical phase retrieval using double deep image priors. arXiv preprint arXiv:2211.00799.

[4] Wang, H., Zhang, X., Li, T., Wan, Y., Chen, T. and Sun, J., 2024. DMPlug: A Plug-in Method for Solving Inverse Problems with Diffusion Models. arXiv preprint arXiv:2405.16749.

---

> ### Author Response · Authors · 2024-12-25
> **Authors' Response to Reviewer YG3X**
>
> We sincerely thank the reviewer for their thoughtful feedback and constructive criticism on our paper. We appreciate the reviewer's recognition of the clarity of our writing, the validity of our core idea, the strength of our empirical results and exhaustive ablation studies. We address the reviewer's concerns point-by-point below. We updated our manuscript as per reviewer’s suggestions (detailed below) and marked the updated portions with **red** fonts in the updated version.
>
> **1\. Comparison to DoubleDIP and Related Works:**
>
> The reviewer correctly points out the relevance of DoubleDIP and subsequent works that utilize multiple deep priors for solving inverse problems. We have added an extended discussion on DoubleDIP and its follow-up works to further clarify the relationship and the differences with our work in the Related Work section (page 4\) in the revised version.
>
> While we acknowledge the conceptual similarity of using multiple priors, we respectfully emphasize the key distinctions between our work and DoubleDIP-based approaches, particularly in the context of intrinsic image decomposition:
>
> * **Suitability of Untrained Priors:** DoubleDIP relies on the self-similarity property of natural images, i.e. the distribution of patches within each separate component is simpler (more uniform) than in the combined image while the patches across the two components are highly dissimilar. This works well when the signal components to be recovered are structurally unaligned. However, the assumption of structural dissimilarity of the components may not hold in the case of IID since albedo and shading are highly aligned or coherent (sharing overall structure, shape, and edges). As shown in the follow-up work UIDNet \[1\], applying DoubleDIP to IID results in cross-contamination and loss of details. UIDNet had to introduce a loss based on a hand-crafted prior, and encoder-decoder network with frequency separated skip connections to mitigate such issues. This highlights that deep untrained priors and internal self-similarity/cross-dissimilarity assumptions may not be well-suited for IID tasks.  We also provided exemplar **qualitative comparisons between doubleDIP and our approach** for the task of IID on both the synthetic and real face images in **Fig. 10 (a)**. It demonstrates that doubleDIP suffers from severe cross-contamination and loss of details, while our approach decomposes the images successfully.
>
> * **Computational Efficiency:** DoubleDIP-based methods, due to their reliance on untrained networks, can be computationally expensive. For example, both DoubleDIP and UIDNet \[1\] report inference times exceeding 10 minutes per image, which is 3-5x slower than our approach.
> * **The Challenge of GAN Inversion:** Unlike DoubleDIP, which uses untrained feedforward networks directly, we employ multiple pre-trained GANs as priors. This necessitates *joint GAN inversion*, a significantly more challenging task than inverting a single GAN, let alone using a simple feedforward prior. Our primary contribution is demonstrating the feasibility and effectiveness of joint GAN inversion for IID, and to the best of our knowledge, ours is the first method to perform it successfully.
>
> We believe that our approach, which extends the core idea of DoubleDIP by leveraging the power of independently trained GAN priors, represents a significant advancement for the specific problem of IID.
>
> *References:*
>
> \[1\]. Zhang, Qing, et al. "Unsupervised intrinsic image decomposition using internal self-similarity cues." IEEE Transactions on Pattern Analysis and Machine Intelligence (2021)
>
> [Cont.]

---

> ### Author Response · Authors · 2024-12-25
> **Authors' Response to Reviewer YG3X (Cont.)**
>
> **2\. Concerns about the kNN Loss:**
>
> We appreciate the reviewer's insightful comments on the kNN loss. We address each concern below:
>
> * **Computational Overhead:** We acknowledge that computing k-nearest neighbors can introduce overhead. However, by leveraging the highly optimized `faiss-gpu` library [2], we have minimized this overhead to less than 15%. Specifically, computing the kNN loss with $k=50$ adds only about 2ms per optimization step per GAN, resulting in a total overhead of only 10 seconds for inverting 3 GANs jointly over 1000 steps (vanilla inversion without the kNN loss takes approximately 73 seconds). Given that the kNN loss operates on the fixed-dimensionality (512) $\mathcal{W}-$space of StyleGAN2, this overhead remains constant regardless of the output resolution or dataset size. We discuss and compare the compute requirements in detail in Sec. 5.5 of our updated manuscript.
> * **Risk of Convergence to Suboptimal Regions:** We respectfully clarify that the kNN loss is not the primary driver of our optimization. The reconstruction loss plays the dominant role in exploring the latent space. The kNN loss acts as a regularizer, ensuring that the optimized latent codes remain within the bounds of their respective GAN's W-space, thus preventing cross-contamination. The balance between global exploration and local confinement is controlled by the $\lambda_{kNN}$ parameter (weightage of kNN loss) set to a low value of 0.0001 in all our experiments, which we found sufficient to maintain the prior while allowing for adequate exploration. We would also like to respectfully correct a factual misunderstanding: we use $k=50$, not $k=100,000$. (In fact, the total number of $w$ vectors we use to represent the entire $\mathcal{W}-$space is $100,000$, out of which only $50$ nearest neighbors are selected at each step to calculate the kNN loss).
>
> * **Marginal Improvements in Table 3:** Table 3 demonstrates the effectiveness of the kNN loss in the context of *single* GAN inversion. While improvements may not be significant, note that the single GAN inversion problem is already reasonably well-solved by standard optimization methods and less susceptible to the exploration vs. local confinement trade-off. The more significant benefits of the kNN loss become evident in the context of *joint* GAN inversion where cross-contamination is a more prevalent issue requiring better balance between exploration vs. local confinement. Substantial improvements brought by kNN loss are demonstrated in Table 4 (for albedo: LPIPS error reduction from 0.120 to 0.104, MSE error reduction from 0.040 to 0.035, and PSNR increase from 22.5 to 23.5) and the qualitative improvements in Figure 11(a).
>
> Further, we provide **additional ablation results in Figure 10(b) comparing kNN loss with in-domain loss**. These results clearly demonstrate the effectiveness of the kNN loss in preventing cross-contamination and preserving fine details during joint inversion, and confirms that it offers right balance between global exploration vs. local confinement without getting stuck in local regions.
>
> *References:*
>
> [2]. Johnson, Jeff, Matthijs Douze, and Hervé Jégou. "Billion-scale similarity search with GPUs." IEEE Transactions on Big Data (2019).
>
> [Cont.]

---

> ### Author Response · Authors · 2024-12-25
> **Authors' Response to Reviewer YG3X (Cont.)**
>
> **3\. Comparison of Memory and Computational Costs:**
>
> A detailed comparison of memory and computational costs with competing methods were already included in Appendix D of our original manuscript. In the revised version, we move it to the main paper in **Sec. 5.5** as requested. We believe our method offers a favorable trade-off between accuracy and efficiency, especially when compared to optimization methods like NextFace or to untrained priors like doubleDIP.
>
> **4\. Justification of GANs over Diffusion Models:**
>
> The reviewer raises a valid point regarding the potential usage of diffusion models as priors. While we acknowledge the impressive results achieved by recent diffusion models in various inverse problems, including the cited work DMPlug, we believe that GANs remain a strong and suitable choice for our task, particularly for IID, for the following reasons:
>
> * **Computational Efficiency:** GANs, especially StyleGAN variants, are generally more compact and computationally efficient than diffusion models, both during training and inference. This is particularly important when training or inferring on multiple models, as in our case. For example, DMPlug requires 635 seconds for inference on a single image as opposed to 190 seconds by our method.
> * **Smooth Latent Space:** The inherent smoothness of GAN latent spaces, especially in StyleGAN, provides useful manipulation capabilities, which is beneficial for tasks like image relighting, as demonstrated in our paper.
> * **Established Inversion Techniques:** The extensive literature on GAN inversion provides a solid foundation for our work, allowing us to leverage established techniques and insights into the properties of GAN latent spaces.
> * **Suitability for IID:** The high degree of alignment between albedo and shading in IID makes it a particularly challenging problem. We have shown that GAN priors, coupled with our joint inversion approach, are effective in this context. On the other hand, DMPlug uses multiple diffusion priors to decompose a blurry image into a clean image, blur kernel, and tilt map – note that the components they recover (image, kernel parameters) are structurally *incoherent* and highly dissimilar from each other, unlike the highly aligned and coherent components like albedo and shading in IID. For example, blur kernel lies in a much lower dimensional space as compared to a clean image, and contains features that are distinctly different from a natural image, making the task easier as compared to separating albedo and shading that shares similar structure, shape, and edges. To this end, adapting diffusion models for IID can be considered an interesting problem for future work.
>
> We agree that exploring diffusion models for IID is a promising direction for future work, especially for handling larger and more diverse datasets. We added a discussion on this in the Related Work and Future Work sections of the revised paper.
>
> **Updates in the manuscript:**
>
> 1. We provided a thorough **comparison of memory and computational costs** with competing methods, including DoubleDIP and relevant GAN/diffusion-based approaches **in Sec. 5.5** of our updated manuscript.
> 2. We **expanded the literature review** to include a more detailed discussion of DoubleDIP, its follow-up works, and relevant works on diffusion model inversion **in Related Work section (Sec. 3)**, highlighting the distinctions and advantages of our approach. We also updated the Future Work section to include ideas regarding diffusion models.
> 3. We provided exemplary **qualitative comparisons** between untrained network-based **doubleDIP and our approach** for the task of IID on both the synthetic and real face images **in Fig. 10 (a)**. It demonstrates that doubleDIP suffers from severe cross-contamination and loss of details, while our approach decomposes the images successfully.
> 4. To further validate the effectiveness of kNN loss, we provided additional **comparisons between results obtained with and without kNN loss in Fig. 10 (b)**, which demonstrates that the kNN loss effectively mitigates the prevalent issues of cross-contamination and loss of details.
>
> We believe that these revisions will further strengthen our paper and address the reviewer's concerns comprehensively. We are grateful for the opportunity to improve our work based on this valuable feedback. We would be happy to answer any further questions.

---

> > ### Comment · Reviewer_YG3X · 2025-01-03
> > **Thanks for the rebuttal**
> >
> > After carefully reviewing the responses, I think that the authors have addressed most of my concerns. Hence, I will update my score and recommend acceptance.

---

### Review · Reviewer_bBJt · 2024-12-05

**Summary Of Contributions:**

This paper addresses Intrinsic Image Decomposition (IID) by leveraging a bank of independently trained GANs for disentangled priors, introducing a Joint GAN Inversion algorithm (JoIN) to decompose images into intrinsic components. The method demonstrates modularity, improved disentanglement, and successful generalization from synthetic to real-world images.

**Audience:**

Yes

**Broader Impact Concerns:**

There is Broader Impact Concerns for this work.

**Claims And Evidence:**

Yes

**Requested Changes:**

1. Revise the claim or provide a clearer explanation for it, specifically addressing why the method, despite also using synthetic data, can better overcome the Sim-to-Real gap.
2. The explanation of the kNN method could be clearer, as mentioned above.
3. If possible, additional experiments and evaluations would be helpful.

**Strengths And Weaknesses:**

**Strengths:**
1. The paper is well-organized with a logical and reasonable framework.
2. The use of independently trained GANs as priors for IID, combined with the joint inversion technique, is innovative and enables simultaneous optimization across multiple GANs.
3. The results in the ablation study demonstrate the effectiveness of the proposed methods.

**Weaknesses:**

1. **Contradictory Claim on Synthetic Data:**
   The authors claim that a key limitation of existing methods is their reliance on synthetic data, leading to poor disentanglement, component cross-contamination, and Sim-to-Real gaps (as mentioned in the abstract). However, their method also heavily relies on synthetic data for training, which undermines the strength of this critique.

2. **Unclear Motivation for kNN Loss:**
   The motivation for the kNN loss could be clarified further. For example, the authors state, "We first sample a large number (100k) of \(z\) codes from a truncated Gaussian distribution to obtain a set of \(w\) codes \(W\)." If the Gaussian distribution refers to the standard Gaussian used in GANs, why does the assumption that "the distribution of \(w\) is isotropic" not hold? A more detailed explanation or empirical evidence would strengthen this argument.

3. **Limited Discussion of Recent Works:**
   The paper lacks references to and comparisons with the latest research. For example, there are no references to works published in 2024 and only a few from 2023. Including comparisons with cutting-edge methods would provide better context for evaluating the contributions.

4. **Insufficient Results on Natural Images:**
   The paper would benefit from more results on natural images, especially quantitative evaluations on real image datasets. For instance, using metrics from recent works such as *[a] Intrinsic Image Decomposition via Ordinal Shading (TOG)* would provide a stronger basis for comparison and highlight the method's performance in real-world scenarios.

---

> ### Author Response · Authors · 2024-12-19
> **Authors' Response to Reviewer bBJt**
>
> We sincerely thank the reviewer for their valuable feedback and insightful comments. Below, we address the concerns raised in detail. We also updated our manuscript and marked the corresponding changes with a **blue** colored text.
>
> **1. Contradictory claim on synthetic data**
>
> We appreciate the reviewer's observation regarding the potentially unclear language used in the abstract to describe the limitations of existing intrinsic image decomposition (IID) methods. We aimed to highlight two key shortcomings of existing image decomposition approaches:
>
> **A. Cross-Contamination:** Most existing learning-based IID approaches train a single network to learn priors for both albedo and shading components simultaneously. This can lead to "cross-contamination," where the signal from one component leaks into the representation of the other during decomposition. To address this, our method employs a distinct strategy: we use a bank of GANs as prior where each GAN is trained independently only on a single component. This separation effectively mitigates cross-contamination. Moreover, since we use a state-of-the-art GAN architecture to learn these priors, we are able to capture each prior more accurately, thus improving the accuracy of decomposition.
>
> **B. The Sim-to-Real Gap:** Existing learning-based priors are trained only on synthetic data since obtaining real data of each intrinsic component is extremely difficult. Such models do not generalize well to real images, particularly because in the case of existing approaches, once the models/priors are trained on synthetic data, they are kept frozen or fixed during inference on real images. Our method, while also utilizing synthetic data for initial training, addresses this limitation through generator fine-tuning: during inference on real images, we allow for careful fine-tuning of our generators, guided by a discriminator loss that acts as a regularizer to preserve the learned priors. This crucial step allows our model to adapt to the specific real image features, effectively bridging the Sim-to-Real gap. By fine-tuning, we avoid the over-reliance on synthetic data leading to significantly improved generalization on real images.
>
> We have revised the introduction and abstract to better articulate these points. We are happy to provide further clarification if needed.
>
> [Continued below]

---

> ### Author Response · Authors · 2024-12-19
> **Authors' Response to Reviewer bBJt (Cont.)**
>
> **2. Unclear motivation for kNN loss**
>
> We appreciate the reviewer's insightful question regarding the non-isotropic nature of the $\mathcal{W}-$space in StyleGAN2. The reviewer correctly points out that the standard Gaussian latent space ($\mathcal{Z}-$space) is isometric. However, note that in our approach we use the intermediate latent space ($\mathcal{W}-$space) for inverting our GANs, and the the subsequent mapping from $\mathcal{Z}-$space to $\mathcal{W}-$space plays a crucial role in StyleGAN2 as explained below.
>
> While the initial $\mathcal{Z}-$space is indeed isotropic (standard Gaussian), StyleGAN2 intentionally introduces a non-linear mapping network (an 8-layer MLP) to transform $\mathcal{Z}$ into the intermediate latent space referred to as $\mathcal{W}-$space. This is a key design element of StyleGAN2, and it is precisely **this non-linear transformation of a mapping network that makes the resulting $\mathcal{W}-$space non-isotropic**. The purpose of this non-linearity is to create a disentangled representation where different dimensions of $\mathcal{W}$ correspond to distinct and independent attributes of the generated images.
>
> As discussed in StyleGAN paper, the $\mathcal{W}-$space is found to better reflect the disentangled nature of the learned distribution [1]. The mapping network effectively warps the isotropic Gaussian distribution to better align with the complex, multi-faceted distribution of real-world image features [1]. This non-isometric nature of $\mathcal{W}-$space is a crucial design choice that enables superior performance, including during GAN inversion, which is why we, like many other inversion methods, perform our optimization in W-space [2,3,4,5].
>
> The original StyleGAN paper [1] provides empirical evidence for this non-isometric property and its benefits in its Figure 6 on page 6. The figure demonstrates how the mapping network transforms isometric Gaussian latent space $\mathcal{Z}$ into an L-shaped non-linear $\mathcal{W}-$space to better capture the underlying structure of the data distribution.
>
> **To this end, we updated our discussion on behavior of $\mathcal{W}-$space in Sec. 3 of our paper. We further provide empirical evidence of non-isometric behavior of $\mathcal{W}-$space in Fig. 3 where we plot the 2D projection of $\mathcal{W}-$space of pre-trained albedo GAN** (on synthetic faces dataset) that shows the distribution varies differently across different dimensions. Using Fig. 3 as reference, we discuss why using kNN loss is more suitable compared to in-domain loss for allowing the exploration while still preserving the priors. We are happy to provide further clarification if needed.
>
> References:
>
> [1]. Karras, Tero et al. “A Style-Based Generator Architecture for Generative Adversarial Networks.” CVPR 2019.
>
> [2]. Xia, Weihao, et al. "GAN Inversion: A survey." IEEE T-PAMI 2022.
>
> [3]. Härkönen, Erik, et al. "Ganspace: Discovering interpretable gan controls." NeurIPS 2020.
>
> [4]. Shen, Yujun, et al. "Interpreting the latent space of gans for semantic face editing." CVPR 2020.
>
> [5]. Wu, Zongze, et al. "Stylespace analysis: Disentangled controls for stylegan image generation." CVPR 2021.
>
> [Continued below]

---

> ### Author Response · Authors · 2024-12-19
> **Authors' Response to Reviewer bBJt (Cont.)**
>
> **3. Discussion on recent works**
>
> We appreciate the reviewer's suggestion to expand our discussion of related work. **In the revised manuscript, we have incorporated discussions of more recent intrinsic decomposition methods at appropriate points within the Introduction and Related Work sections.** Specifically, we have included a discussion of feed-forward approaches that predict either albedo alone [6, 8] or both albedo and shading [7]. We also added a reference to a recent review paper on IID [9] in our discussion. Furthermore, we have added a discussion on a recent data-driven, end-to-end image relighting method Latent Intrinsics [10] (published in NeurIPS 2024) that predicts albedo as a byproduct. This being the latest work (published in NeurIPS, 2024) in the area, we have also added a detailed comparison with this method in our results, providing both qualitative and quantitative evaluations in Figure 7 and Table 2, respectively.
>
> **4. Additional Results and Comparisons**
>
> Following the reviewer's suggestion, we have expanded our evaluation to include additional results on natural images. Specifically, **we present new results on natural images of faces in Figure 7 (a) and (b)**. Furthermore, the appendix now includes supplementary results on natural images from both the materials dataset (Figure 16) and the faces dataset (Figures 19 and 20). **We also provide a new quantitative comparison with the recently published Latent Intrinsics method (NeurIPS 2024) [10]. This comparison, illustrated in Figure 7 (b) for real faces and Table 2 for synthetic faces**, demonstrates that our method achieves superior albedo estimation accuracy compared to Latent Intrinsics.
>
> *References:*
>
> [6]. Jin, Yeying, et al. "Estimating reflectance layer from a single image: Integrating reflectance guidance and shadow/specular aware learning." AAAI 2023.
>
> [7]. Careaga, Chris, and Yağız Aksoy. “Intrinsic Image Decomposition via Ordinal Shading.” ACM Transactions on Graphics, 2023.
>
> [8]. Luo, Jundan, et al. "CRefNet: Learning Consistent Reflectance Estimation With a Decoder-Sharing Transformer." IEEE Transactions on Visualization and Computer Graphics, 2023.
>
> [9]. Liu, Siyuan et al. “A Review of Intrinsic Image Decomposition.” 2024 3rd International Conference on Image Processing and Media Computing (ICIPMC) (2024).
>
> [10]. Zhang, Xiao et al. “Latent Intrinsics Emerge from Training to Relight.” NeurIPS, 2024.
>
>
> **Addressing Broader Impact Concerns**
>
> In Appendix Section E, we discuss the broader impact concerns of our work, particularly regarding the use of human data and the ethical implications of our approach. To summarize, we only use human face images (specifically from the FFHQ dataset) for evaluation purposes, strictly adhering to its usage guidelines. All our models are trained exclusively on synthetic data. Although our method focuses on intrinsic decomposition and relighting, and does not alter facial appearance, we acknowledge potential risks such as misrepresentation of context, reinforcement of biases inherent in the training data, and challenges for image authentication. Therefore, we advocate for responsible development and deployment of this technology alongside further research into mitigating these risks.
>
> We are happy to address any further questions by the reviewer. Thanks!

---

> > ### Comment · Reviewer_bBJt · 2025-01-04
> > **Thanks for the reponse.**
> >
> > Thanks for the detailed response. I do not have further questions.
> > I tend to accept the paper.

---

### Author Response · Authors · 2024-12-27
**General response to the reviewers**

Dear Reviewers,

Thank you for reviewing our manuscript and providing valuable feedback. We have addressed your comments and revised the manuscript accordingly. Responses to specific points are detailed in our replies to your reviews.

An updated manuscript is now available. Revisions are outlined in our individual responses. The updates addressing the concerns of the reviewer bBJt and reviewer YG3X are marked with **blue** and **red** colored texts respectively.

We are available to answer any further questions.

Sincerely,

The Authors

---

### Author Response · Authors · 2025-02-12
**Thank you to the Action Editor and Reviewers**

The authors extend their sincere gratitude to the action editor and reviewers for their constructive feedback and insightful suggestions, which have contributed substantially to the improvement of this work. We submitted the camera-ready version that incorporates the suggestions provided by the reviewers in their reviews.

---

### Decision · Action_Editor_ruC9 · 2025-01-13

**Recommendation:** Accept as is

**Comment:**

The submission studies intrinsic image decomposition and proposes jointly inverting the latent codes of a set of GANs and combining their outputs to reproduce the input.  The idea makes sense and is supported by experimental results.  Reviewers like the clear formulation and presentation and the empirical results.  Meanwhile, they have also pointed out that similar ideas have been explored in the literature.  Based on the TMLR criteria, I believe this submission can be accepted.

**Audience:**

Yes

**Claims And Evidence:**

Yes, the claim on using different priors for components in Intrinsic Image Decomposition (IID) has been supported by experimental results.